# On the Stability and Convergence of Robust Adversarial Reinforcement Learning: A Case Study on Linear Quadratic Systems

**Kaiqing Zhang**
ECE and CSL
University of Illinois at Urbana-Champaign
kzhang66@illinois.edu

**Bin Hu**
ECE and CSL
University of Illinois at Urbana-Champaign
binhu7@illinois.edu

**Tamer Başar**
ECE and CSL
University of Illinois at Urbana-Champaign
basar1@illinois.edu

## Abstract

Reinforcement learning (RL) algorithms can fail to generalize due to the gap between the simulation and the real world. One standard remedy is to use *robust adversarial RL* (RARL) that accounts for this gap during the policy training, by modeling the gap as an adversary against the training agent. In this work, we reexamine the effectiveness of RARL under a fundamental robust control setting: the *linear quadratic* (LQ) case. We first observe that the popular RARL scheme that greedily alternates agents' updates can easily *destabilize the system*. Motivated by this, we propose several other policy-based RARL algorithms whose convergence behaviors are then studied both empirically and theoretically. We find: i) the conventional RARL framework (Pinto et al., 2017) can learn a destabilizing policy if the initial policy does not enjoy the *robust stability* property against the adversary; and ii) with robustly stabilizing initializations, our proposed double-loop RARL algorithm provably converges to the global optimal cost while maintaining robust stability on-the-fly. We also examine the stability and convergence issues of other variants of policy-based RARL algorithms, and then discuss several ways to learn robustly stabilizing initializations. From a robust control perspective, we aim to provide some new and critical angles about RARL, by identifying and addressing the stability issues in this fundamental LQ setting in continuous control. Our results make an initial attempt toward better theoretical understandings of policy-based RARL, the core approach in Pinto et al., 2017.

## 1 Introduction

Reinforcement learning (RL) can fail to generalize due to the gap between the simulation and the real world. One common remedy for this is to use *robust adversarial RL* (RARL) that accounts for this gap during the policy training, by modeling the gap as an adversary against the training agent [1, 2]. To achieve the goal of learning a policy that is robust against a family of possible model uncertainties, RARL jointly trains a *protagonist* and an *adversary*, where the protagonist learns to robustly perform the control tasks under the possible disturbances generated by its adversary. Despite the recent development of various robust RL algorithms, especially for continuous control tasks [2, 3, 4, 5, 6], there is no clean baseline delineating the robustness of the policies learned by such a framework.

Motivated by the deep connection between RARL and robust control theory, this paper reexamines the effectiveness of RARL under a fundamental robust control setting: the linear quadratic (LQ) case. Specifically, we consider a RARL setting where the state transition follows linear dynamics, and the reward/cost of the protagonist and the adversary is quadratic in the state and the joint control actions. Such a linear quadratic setting is one of the most fundamental models for robust control [7, 8], and can be viewed as the robust version of the classic linear quadratic regulator (LQR) model, one of the most fundamental models in continuous control and RL [9, 10, 11, 12]. Such a model also fits within the general RARL proposed in the pioneering work [2], with continuous state-action spaces.

The popular RARL scheme with a *policy optimization* framework [2], though enjoying great empirical successes, has not yet been put on a solid theoretical footing. Some pressing issues are whether/where the policy-based RARL scheme converges, what robust performance can be guaranteed, and whether it preserves certain robustness during learning. None of these issues have been rigorously resolved in RARL. Nonetheless, these issues are of paramount importance when applying RARL to control systems, especially safety-critical ones. A non-convergent algorithm and/or a failure to preserve robustness or even stability during learning, i.e., *robust/stable on-the-fly*, can cause detrimental consequences to the system. A destabilized system cannot be used for learning anymore, as the objective is then not even well-defined. See more discussions in §3. In this work, we make an attempt toward addressing these questions, by reexamining RARL in the LQ setup. Inspired by the recent results on *policy-based* RL for LQR [10, 13, 14, 15, 16, 17, 18] and related variants [19, 20, 21], we develop new theoretical results on the stability and convergence of LQ RARL.

In this paper, we first observe some negative results by applying the popular policy-based RARL scheme from [2] onto the LQ setup: the alternating update as in [2] can easily destabilize the system. We identify that guaranteeing stability during learning is non-trivial, which critically relies on both the *update rule* and the *controller*[1] *initialization*. Some seemingly reasonable initializations, e.g., simply a stabilizing controller, can still fail. Motivated by this, we develop an *update-initialization* pair that provably guarantees both *robust stability* and *convergence*. It seems that both the stability issues (negative results) and the significance of robust stability (positive results) have been overlooked in the RARL literature, if they were to be applied in continuous control tasks. We highlight our contributions as follows:

- We identify several stability issues of the popular RARL scheme in the LQ setup, showing that guaranteeing robust stability during learning requires a non-trivial intertwinement of update rules and controller initializations.
- We propose a double-loop natural policy gradient (PG) algorithm which updates the protagonist's policy incrementally. We prove that this algorithm, with some robust-control meaningful initialization, is guaranteed to maintain robust stability *on-the-fly* and leads to convergence to the optimal cost. We also explore the potential stability and convergence issues of several other algorithm variants.
- We develop new robustification techniques, from an $\mathcal{H}_\infty$-robust control perspective, to learn such robustly stabilizing initializations, with empirical validations.

We expect that both our theoretical and experimental findings will shed new lights on RARL, from a rigorous robust control perspective. We also note that although our system dynamics are linear, our model can handle general *nonlinear* disturbances, which can be viewed as the approximation error of linearizing nonlinear dynamics.

**Related work.** Exploiting an adversary to tackle model-uncertainty and improve sim-to-real performance in RL dates back to [1], which, interestingly, stemmed from the $\mathcal{H}_\infty$-robust control theory. Actor-critic robust RL algorithms were proposed therein, though without theoretical guarantees for either convergence or stability. This minimax idea was then carried forward in the popular RARL scheme [2], with great empirical successes, which has then been followed up and improved in [22, 23]. The policy-based RARL algorithms therein serve as the starting point for our work. Following the same worst-case modeling idea for uncertainty, robust RL has also been investigated in the realm of robust Markov decision processes (MDPs) [24, 25, 26]. Our LQ RARL setup can be viewed as a specification of robust MDP in the continuous control context. Other recent advances on robust RL for continuous control include [4, 5, 27, 6]. An increasing attention has also been paid to ensuring robustness and stability in general data-driven control [28, 29, 30].

Our LQ RARL model has a strong connection to LQ zero-sum dynamic games [31], due to the significant role it plays in $\mathcal{H}_\infty$-control [32, 7]. Recently, several provably convergent policy-based RL methods have been developed for this zero-sum setting [33, 34, 35, 36], together with some negative results for general-sum LQ games [37]. In [33], the first convergence study of direct policy search for LQ games was established with a projection operation on the iterates, which however can be restrictive, with only few robust control implications. In [34], the projection was removed through the lens of robust control, and the results in [33] were improved based on an implicit regularization argument, showing that certain policy-based algorithms can preserve robustness during iterations automatically. An independent work [35] then also removed the projection, with a different proof technique and under different assumptions. In this aspect, our analysis in the present work mainly relies on our own techniques developed in [34], and our assumptions align with the standard ones in the robust control literature. More recently, approximate policy iteration algorithms have also been proposed for LQ zero-sum games with stochastic parameters [36].

## 2 Preliminaries

### 2.1 LQ RARL and Robust Control

We first introduce some background on LQ RARL and its close connection to robust control. Consider a linear dynamical system

$$x_{t+1} = Ax_t + Bu_t + Cw_t, \tag{2.1}$$

where the system state is $x_t \in \mathbb{R}^d$, the control input of the agent at time $t$ is $u_t \in \mathbb{R}^{m_1}$, the disturbance or any unmodeled error at time $t$ is denoted by $w_t \in \mathbb{R}^{m_2}$. The matrices satisfy $A \in \mathbb{R}^{d \times d}$, $B \in \mathbb{R}^{d \times m_1}$, and $C \in \mathbb{R}^{d \times m_2}$. We define the one-stage cost as $c_t(x_t, u_t, w_t) := x_t^\top Q x_t + u_t^\top R^u u_t - w_t^\top R^w w_t$ with some positive definite matrices $Q \in \mathbb{R}^{d \times d}$, $R^u \in \mathbb{R}^{m_1 \times m_1}$, and $R^w \in \mathbb{R}^{m_2 \times m_2}$. Then, the objective of the learning agent is to find $\{u_t\}_{t \geq 0}$ to minimize an accumulative cost subject to the worst-case disturbance:

$$\min_{\{u_t\}_{t \geq 0}} \sup_{\{w_t\}_{t \geq 0}} \mathcal{C}(\{u_t\}_{t \geq 0}, \{w_t\}_{t \geq 0}) := \mathbb{E}\left[ \sum_{t=0}^{\infty} c_t(x_t, u_t, w_t) \right], \tag{2.2}$$

where the expectation is taken over the trajectory $\{x_t\}_{t \geq 0}$. For simplicity, we assume that the only randomness stems from the initial state $x_0 \sim \mathcal{D}$, with some distribution $\mathcal{D}$ and $\mathbb{E}[x_0 x_0^\top] = \Sigma_0 > 0$.

**Remark 2.1** (LQ RARL Model). The model introduced above has been widely used in the robust control literature [7, 8] to handle adversary/uncertainty in continuous control. It mirrors the standard (finite) robust MDP setting for robust RL [38, 24, 25], where both the reward and the transition model have some uncertainty, and a *minimax* formulation was also developed. In the LQ model, such uncertainty also enters *both* the cost and the dynamics, through the disturbance $w_t$. As such, this LQ dynamic game can be viewed as a robust MDP substantiated in the continuous control context. Such a minimax formulation has also been adopted in the RARL work [2], by treating $\{u_t\}_{t \geq 0}$ and $\{w_t\}_{t \geq 0}$ as the protagonist's actions and adversary's attacks, respectively. Finally, since $w_t$ is allowed to be a nonlinear feedback function of $x_t$, we can also view (2.1) as a linear approximation of some *nonlinear* model, with $Cw_t$ capturing the model approximation error. A desired controller should be robust to this modeling error.

The formulation in (2.2) naturally leads to a *zero-sum dynamic game*, where the disturbance is viewed as another player of the game besides the learning agent. At the *Nash equilibrium* (NE), the solution concept of the game, a pair of control-disturbance sequences $\{u_t^*\}_{t \geq 0}$ and $\{w_t^*\}_{t \geq 0}$ satisfies

$$\mathcal{C}(\{u_t^*\}_{t \geq 0}, \{w_t\}_{t \geq 0}) \leq \mathcal{C}(\{u_t^*\}_{t \geq 0}, \{w_t^*\}_{t \geq 0}) \leq \mathcal{C}(\{u_t\}_{t \geq 0}, \{w_t^*\}_{t \geq 0}) \tag{2.3}$$

for any $\{u_t\}_{t \geq 0}$ and $\{w_t\}_{t \geq 0}$. The NE control sequence $\{u_t^*\}_{t \geq 0}$ is *robust*, in that it minimizes the cost against *any* causal worst-case disturbance. This minimax robustness argument is also used in robust RL in the model of robust MDPs [24, 25] and RARL [2, 22, 23].

As such, it suffices to find the NE of the game, in order to obtain a robust controller. It is known that the NE of the LQ game (2.2) can be attained by state-feedback controllers under standard assumptions [7] (see Assumption A.1, which is made throughout this paper), i.e., there exists a pair of matrices $(K^*, L^*) \in \mathbb{R}^{m_1 \times d} \times \mathbb{R}^{m_2 \times d}$, such that $u_t^* = -K^* x_t$ and $w_t^* = -L^* x_t$. Hence, it suffices to search over the *stabilizing* control gain matrices $(K, L)$ (policy parameters), for such NE. This naturally

motivates the use of policy-based RARL schemes. Finally, we would like to point out that although one NE disturbance $w_t^* = -L^* x_t$ is in linear state-feedback form, the NE controller $\{u_t^*\}_{t \geq 0}$ can tolerate even *nonlinear* disturbances [7]. Due to space limitation, more intuition on the robustness of $\{u_t^*\}_{t \geq 0}$, and more background on the solution to (2.2), are deferred to §A.

## 2.2 Policy-Based LQ RARL Scheme

As mentioned above, policy-based RARL on the parameter pair $(K, L) \in \mathbb{R}^{m_1 \times d} \times \mathbb{R}^{m_2 \times d}$ can solve (2.2). Indeed, such policy-based approaches have been the core of the popular RARL scheme in [2]. Thus, (2.2) can be equivalently re-written as

$$\min_K \max_L \ \mathcal{C}(K, L) \tag{2.4}$$

where the solution NE $(K^*, L^*)$ satisfies $\mathcal{C}(K^*, L) \leq \mathcal{C}(K^*, L^*) \leq \mathcal{C}(K, L^*)$, and the cost $\mathcal{C}(K, L) = \mathbb{E}_{x_0 \sim \mathcal{D}}\big(x_0^\top P_{K,L} x_0\big)$, for any *stablilizing* policy pair $(K, L)$ with $\rho(A - BK - CL) < 1$ [33, 35], when $u_t = -Kx_t$ and $w_t = -Lx_t$ are substituted in. $P_{K,L}$ here is the unique solution to the Lyapunov equation

$$P_{K,L} = Q + K^\top R^u K - L^\top R^w L + (A - BK - CL)^\top P_{K,L}(A - BK - CL). \tag{2.5}$$

The policy gradient of $\mathcal{C}(K, L)$ with respect to $K$ and $L$ can be obtained using $P_{K,L}$ [33, 35]. More significantly, under standard conditions, the objective $\mathcal{C}(K, L)$, though nonconvex in $K$ and nonconcave in $L$, has a nice property that *all stationary-points are NE*. This justifies the use of policy-based RARL in this setup, as finding first-order stationary-point is sufficient. We summarize the PG formulas and landscape results in Lemmas A.3 and A.5 in §A.3.

Now it is tempting to follow the original RARL scheme in [2]. Specifically, it alternates between the two players: the adversary improves its disturbing policy $L$ with the agent's policy $K$ fixed; the agent then learns its policy $K$ with a fixed $L$. This sequence is repeated until convergence (if they do). We summarize such a RARL scheme in Algorithm 1 in §A for completeness. We abstract out the update rule for $K_n$ and $L_n$ as *PolicyOptimizer* functions, which can be policy gradient, or natural PG updates that have been widely used in the LQ setup [10, 15, 33, 35]. Specifically, one can substantiate the *PolicyOptimizer* as

$$L' = \left\{ \begin{array}{ll} L + \eta \nabla_L \mathcal{C}(K, L) \Sigma_{K,L}^{-1} & \text{if NPG} \\ L + \eta \nabla_L \mathcal{C}(K, L) & \text{if PG} \end{array} \right. ; \qquad K' = \left\{ \begin{array}{ll} K - \eta \nabla_K \mathcal{C}(K, L) \Sigma_{K,L}^{-1} & \text{if NPG} \\ K - \eta \nabla_K \mathcal{C}(K, L) & \text{if PG} \end{array} \right. ,$$

where $\eta > 0$ is the stepsize and $\Sigma_{K,L} := \mathbb{E}_{x_0 \sim \mathcal{D}} \sum_{t=0}^{\infty} x_t x_t^\top$ is the state correlation matrix. $\nabla_L \mathcal{C}(K, L)$, $\nabla_K \mathcal{C}(K, L)$, and $\Sigma_{K,L}$ can be estimated via sample trajectories, as in LQR [10, 39].

## 3 Stability Issues in Policy-Based LQ RARL

We first identify several stability issues of the RARL scheme in [2] in this LQ setup.

**Remark 3.1** (Significance of Stability in Learning)**.** Stability is a crucial property required for learning-based control algorithms. Particularly, if the algorithm destabilizes the system during learning, some catastrophic and irreversible consequences will be caused to the system. Moreover, the cost to be minimized (cf. (2.2)) is not even well-defined, and thus the learning process cannot proceed. Such an issue, if it exits, would only be worsened in sample-based learning, as the stochasticity in the data brings in more instability. To better illustrate the issues, we focus here on an ideal case, where the exact policy gradient is available. We show that even this ideal case will cause stability issues.

### 3.1 Stability Issue due to Bad Initialization

At a first glance, any policy pair $(K_0, L_0)$ that stabilizes the system (2.1) can be a reasonable initialization for Algorithm 1. However, such a naive initialization can cause severe instability.

**Thought Experiment.** It is reasonable to start with some $K_0$ that stabilizes the system (2.1) when there is *no* disturbance. Then $(K, 0)$ stabilizes the system (2.1), and one can try to obtain $L_1$ by applying $L' = \text{PolicyOptimizer}(K_0, L)$ with an initialization $L = 0$. Suppose the policy optimizer works well and eventually leads to an exact solution $L(K_0) \in \text{argmax}_L \mathcal{C}(K_0, L)$ (if it exists). A

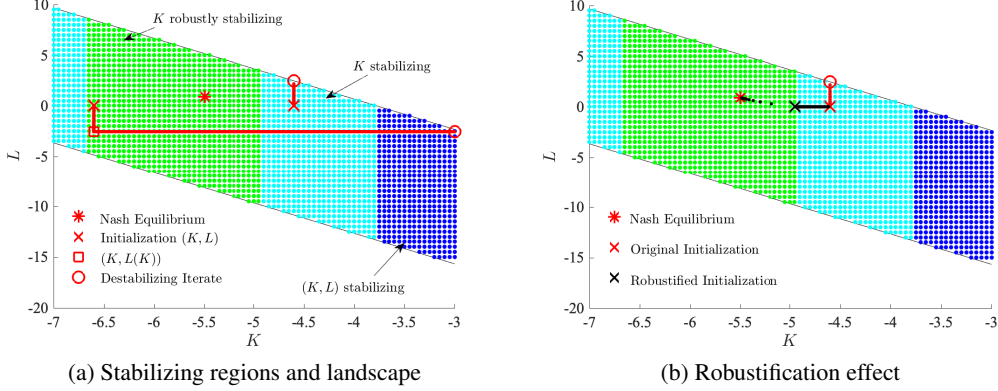

(a) Stabilizing regions and landscape      (b) Robustification effect

Figure 1: (a) Different stabilizing regions of the policy pair $(K, L)$ for **Examples** 3.2 and 3.6. The blue, cyan, and green regions represent the stabilizing region of the pair $(K, L)$, the stabilizing region of $K$, and the robustly stabilizing region of $K$, respectively. (b) Effectiveness of our robustification technique. For the fixed initialization $K_0$ that is not robustly stabilizing (but stabilizing), updating $L$ will destabilize the system (see also in **Example** 3.2). After robustifying it to the green region, even the simultaneous-descent-ascent updates of $K$ and $L$ in §4.4 converge to the NE successfully.

key question is whether $(K_0, L(K_0))$ still stabilizes the system (2.1). Unfortunately, if we only know $\rho(A - BK_0) < 1$, we cannot guarantee $\rho(A - BK_0 - CL(K_0)) < 1$. The fundamental issue here is that for LQ games, the cost $\mathcal{C}(K, L)$ may be finite even for some non-stabilizing $(K, L)$ (i.e. $\rho(A - BK - CL) \geq 1$). Hence the existence of $L(K)$ does not mean $\rho(A - BK - CL(K)) < 1$. See Section 3.1 in [35] for more discussions. We thus may need some extra condition on $K$, besides being stabilizing, in order to guarantee that $(K, L(K))$ is also stabilizing. Now we present a simple example to demonstrate this stability issue due to a bad choice of $K_0$.

**Example 3.2.** Consider $A = 2.7$, $B = -0.45$, $C = -0.15$, $R^u = 1$, $R^w = 2.0736$, $Q = 1$. The NE $(K^*, L^*) = (-5.4957, 0.8834)$. As shown in Figure 1 (a), the blue region is the one where the pair $(K, L)$ is stabilizing, i.e., $\rho(A - BK - CL) < 1$; the cyan one is the stabilizing region of $K$, i.e., $\rho(A - BK) < 1$, that intersects with the blue one, which is smaller and contains the NE control gain $K^*$. As shown in Figure 1 (a), if we start the conventional RARL algorithm from $(K_0, L_0) = (-4.6, 0)$, which stabilizes the system, and improve $L_0$ with $K_0$ fixed, the trajectory still goes up straight to the destabilizing region. The update of $L$ follows NPG with stepsize $\eta = 10^{-10}$. In fact, for fixed $K_0$, the inner-loop problem $L_1 = \mathrm{argmax}_L \mathcal{C}(K_0, L)$ *does not* even have a solution, as the induced Riccati equation cannot be solved (`Matlab` also returns a solvability issue).

To impose an additional condition on $K$, we first introduce the following Riccati equation

$$P_K^* = Q + K^\top R^u K + (A - BK)^\top \widetilde{P}_K^* (A - BK), \tag{3.1}$$

where $P_K^* = P_{K,L(K)}$, and $\widetilde{P}_K^* = P_K^* + P_K^* C (R^w - C^\top P_K^* C)^{-1} C^\top P_K^*$ if $R^w - C^\top P_K^* C > 0$. Then, we introduce the following condition on $K$.

**Definition 3.3** (Robust Stability Condition for $K$). First, $K$ is stabilizing, i.e., $\rho(A - BK) < 1$. Also, the Riccati equation (3.1) admits a minimal positive semidefinite solution $P_K^* \geq 0$ such that $R^w - C^\top P_K^* C > 0$.

One key consequence of $K$'s robust stability is the following lemma, whose proof is given in §A.4.1.

**Lemma 3.4.** For any $K$ satisfying the robust stability condition, the inner-loop problem $\max_L \mathcal{C}(K, L)$ is well-defined (admits a finite optimal value) and admits a *unique* stabilizing solution among all stabilizing $L$, given by $L(K) = (-R^w + C^\top P_K^* C)^{-1} C^\top P_K^* (A - BK)$.

The robust stability condition is in fact not only *sufficient* for the inner-loop problem to be well-defined, but also *almost necessary*, see [7, Lemma 3.6] for more discussions. Moreover, it also has many important robust control implications. One can roughly think of that the robust stability condition enforces the controller $K$ to robustly stabilize the system subject to the worst case attack $L(K)$. More formal explanations on how this condition induces robustness to the control design are

provided in §A.4. This abstract condition can be equivalently transformed into a more checkable form using the Bounded Real Lemma [7, 8, 40]. We review this result here. Given $(A, B, C, Q, R^u, R^w)$, we define a matrix function

$$M(P, K) := \begin{bmatrix} Q + K^\top R^u K & 0 \\ 0 & -R^w \end{bmatrix} + \begin{bmatrix} (A - BK)^\top P(A - BK) - P & (A - BK)^\top PC \\ C^\top P(A - BK) & C^\top PC \end{bmatrix},$$

where $P$ and $K$ are matrices with compatible dimensions.

**Lemma 3.5** (Bounded Real Lemma [41, 8])**.** The robust stability condition (Definition 3.3) holds if and only if there exists some $P > 0$, such that $M(P, K) < 0$.

Based on the above result, we can check the robust stability condition for any given $K$ by testing a semidefinite program $M(P, K) < 0$. This will be useful in the ensuing analysis.

### 3.2 Stability Issue due to Bad Choices of $(N_K, N_L)$

A less careful choice of $N_K, N_L$ may also destabilize the system. Suppose that both $N_K$ and $N_L$ are large such that $L(K)$ and $K(L)$ are accurately computed at each iteration. Algorithm 1 thus iterates as $L' = L(K)$ and $K' = K(L')$. Then the LQ RARL algorithm becomes the *best-response* algorithm [42], known to be not always convergent in game theory. We show next that this can easily happen in LQ RARL, even if the initialization $K_0$ is robustly stabilizing.

**Example 3.6.** Consider the same system as in **Example** 3.2. Let the green region denote the robustly stabilizing region of $K$. We start with a robustly stabilizing $K_0 = -6.6$, and let $L_0 = 0$. We then use RARL with NPG update with stepsize $\eta = 0.005$. We choose $N_K = N_L = 100$. First, for fixed $K_0$, the NPG update for $L$ easily converges to the solution $L(K_0) = -2.5606$ (the red square in Figure 1 (a)) within $N_L$ iterations. Recall that the existence and stability of such an $L(K_0)$ are guaranteed by the robust stability of $K_0$. However, if we then continue to fix $L_1 = L(K_0)$, and improve $K$, even with stepsize $10^{-10}$ (infinitesimal), it will still go right straight to the destabilizing region, if it updates long enough. This phenomenon is essentially due to that for fixed $L_1 = L(K_0)$, the inner-loop problem $\min_K \mathcal{C}(K, L_1)$ may not in turn necessarily be well-defined (admit a finite optimal value). Hence, even with a robustly stabilizing initialization $K_0$, the RARL update can still destabilize the system easily if $N_K$ and $N_L$ are not set properly.

The stability issues above demonstrate the significance of both the initialization (a robustly stabilizing $K_0$) and the update rule (properly chosen $(N_K, N_L)$), in developing policy-based LQ RARL algorithms. Next, we introduce such an *update-initialization* pair that is provably stable and convergent.

## 4 Algorithms and Theory

### 4.1 A Double-Loop Algorithm

In this section, we present a specific double-loop algorithm with both *stability* and *convergence* guarantees. This algorithm uses an outer-loop NPG update to improve the agent's policy $K$ by

$$K_{n+1} = K_n - \eta \nabla_K \mathcal{C}(K_n, L(K_n)) \Sigma_{K_n, L(K_n)}^{-1} \tag{4.1}$$

where $L(K_n) := \operatorname{argmax}_L \mathcal{C}(K_n, L)$ is solved within an inner loop with fixed $K_n$. For each inner loop, the policy for $L$ is always initialized so that $(K_n, L)$ is stabilizing ($L = 0$ will suffice), and the method used in the inner loop is NPG. Ideally, this algorithm can be viewed as a special case of Algorithm 1 with $N_K = 1$ and $N_L \to \infty$. In this section, we show that this algorithm can guarantee both the stability of the policy pair $(K, L)$ and the robust stability of $K$, along the optimization process, and provably converges to $(K^*, L^*)$ if initialized at a robustly stabilizing policy $K_0$. We also provide simulations to support our theory, mostly deferred to §C due to space limitation.

### 4.2 Outer Loop Analysis

We first show that the outer-loop iterate $K_n$ of our algorithm is guaranteed to satisfy the robust stability condition, if $K_0$ is robustly stabilizing.

**Lemma 4.1** (Preserving Robust Stability)**.** For any $K_n$ satisfying the robust stability condition (Definition 3.3), suppose that the stepsize $\eta$ satisfies $\eta \le 1/(2\|R^u + B^\top \widetilde{P}_{K_n}^* B\|)$. Then, $K_{n+1}$ obtained from (4.1) also satisfies the robust stability condition.

Now we can prove the main convergence result, showing that the outer-loop update (4.1) is guaranteed to find the Nash equilibrium control gain $K^*$ with a sublinear rate.

**Theorem 4.2.** Suppose that $K_0$ satisfies the robust stability condition (Definition 3.3). With the stepsize $\eta \leq 1/(2\|R^u + B^\top \widetilde{P}^*_{K_0} B\|)$, the update (4.1) converges to $K^*$ at a sublinear rate $O(1/N)$ .

**Remark 4.3** (Local Results with Faster Rates). In addition to the global sublinear convergence, one can further obtain local faster rates for (4.1) using the so-called *gradient dominance* property [43, 44] that has been proved to hold locally for zero-sum LQ games [33, 34]. It is even possible to prove a local superlinear rate if we replace the NPG update in (4.1) with the so-called Gauss-Newton update and set learning rate as $\eta = 1/2$. Due to space limitation, we defer more discussions on the Gauss-Newton update to the appendix.

## 4.3 Inner Loop Analysis

For each robustly stabilizing $K_n$, our algorithm solves an inner-loop problem to obtain $L(K_n) := \operatorname{argmax}_L \mathcal{C}(K_n, L)$. The inner-loop algorithm is initialized at some stabilizing $L$ ($L = 0$ suffices) and applies the NPG update as follows

$$L' = L + \eta_L \nabla_L \mathcal{C}(K_n, L) \Sigma^{-1}_{K_n, L}. \tag{4.2}$$

The inner-loop problem is essentially a non-standard LQR problem with *indefinite* cost weighting matrix $-Q - K^\top R^u K$ [33, 35]. The coercivity property used in [10] does not hold and a separate stability analysis for the iterate $L$ during optimization is needed. Motivated by a recent novel contradiction argument in [35], we show in the following lemma that the NPG update (4.2) for $L$ can guarantee both stability and convergence.

**Lemma 4.4.** Suppose $K_n$ satisfies the robust stability condition (Definition 3.3). Initialize the inner-loop optimization at a stabilizing $L$. With stepsize $\eta_L \leq 1/(2\|R^w - C^\top P_{K_n, L} C\|)$, the inner-loop NPG update (4.2) is guaranteed to be stabilizing and converges to $L(K_n)$ at a linear rate.

The above lemma is similar to the results in [35], but the adopted assumptions are quite different. Our assumption aligns with the standard ones in the robust control theory literature. Specifically, our condition on $K$, i.e., the robust stability condition, which has some robust control implications and eventually leads to the $\mathcal{H}_\infty$-based initialization technique presented in the next section, is different from the condition required in [35]. The proof of Lemma 4.4 is included in §B.3 for completeness. Finally, we note that using a similar but more involving argument, the PG update of $L$ can also be shown to converge at a linear rate. We refer interested readers to [35, Theorem 7.4] for more details. We also provide simulations in §C.2 to validate our theory. As expected, the double-loop algorithm converges to the NE control gain $K^*$ successfully.

## 4.4 Other Variants with Possible Stability & Convergence Issues

Intuitively, one does not need to solve the inner-loop optimization exactly. We have implemented the algorithm with different values of $N_L$. The numerical results indicate that the algorithms even with $N_L = 1$ work well if the initial policy $K_0$ satisfies the robust stability condition. We have also identified examples where these algorithms fail to converge when the initial policy does not satisfy the robust stability condition. However, there are also cases with a finite $N_L > 1$, when a robustly stabilizing $K_0$ fails to lead to convergence and/or leads the iterates to remain robustly stabilizing. These interesting findings reaffirm the complicated intertwinement between *update rule* and *initialization*, in order to guarantee the stability and convergence of LQ RARL in general. This in turn reflects the significance of our results in §4.1-§4.3, as we have provided an update-initialization pair that provably works.

Besides the case with $N_K = N_L = 1$, we further test the performance of the NPG descent-ascent algorithm that updates $(K, L)$ simultaneously, with even an identical stepsize. The update rule is

$$K' = K - \eta \nabla_K \mathcal{C}(K, L) \Sigma^{-1}_{K, L}, \qquad L' = L + \eta \nabla_L \mathcal{C}(K, L) \Sigma^{-1}_{K, L},$$

which is easier to implement than the double-loop/two-timescale update in practice. Surprisingly, this algorithm also works well under our robustly stabilizing initialization. Due to space limitation, we briefly summarize our key findings here and defer the detailed simulation studies to §C. Based on our observations, the descent-ascent updates with properly chosen $N_K$ and $N_L$ are effective,

i.e., converge to the NE successfully. Most of the successful cases require the initialization $K_0$ to satisfy the robust stability condition. On the other hand, if the initialization is not robustly stabilizing, descent-ascent updates do not always work (but may work in some cases). These observations reinforce the significance of a *robustly stabilizing initialization*. It remains unclear as to exactly what type of initial conditions would be needed for the descent-ascent methods.

## 5   Robustify Initializations via $\mathcal{H}_\infty$-Approach

As shown earlier, if one only requires the initial policy $K$ to be stabilizing, i.e., stabilize the system for $L = 0$, the policy-based LQ RARL algorithms can fail. On the other hand, the robust stability condition on $K$ is provably significant for the double-loop algorithm, and empirically also useful for other variants such as alternating or multi-step update rules. Hence, it is imperative to initialize $K$ to be robustly stabilizing. To this end, we propose a robustification technique, which aims to robustify any *stabilizing* control gain to be a *robustly stabilizing* one, from an $\mathcal{H}_\infty$-control perspective. We first introduce some notations from control theory. Let $G := \left[ \begin{array}{c|c} A & B \\ \hline C & \mathbf{0} \end{array} \right]$ denote the state-space realization of the model $x_{t+1} = Ax_t + Bw_t$, and $z_t = Cx_t$. Then, the $\mathcal{H}_\infty$-norm of $G$ is defined as

$$\|G\|_\infty = \sup_{\theta \in [0,2\pi)} \lambda_{\max}^{1/2}[B^\top (e^{-j\theta}I - A)^{-\top} C^\top C(e^{j\theta}I - A)^{-1}B].$$

Given $(A, B, C)$, the $\mathcal{H}_\infty$-norm can be efficiently calculated using the Hamiltonian bisection method [45]. The `Matlab` function `hinfnorm` can be directly called for this calculation. When $(A, B, C)$ are unavailable, the $\mathcal{H}_\infty$-norm can also be efficiently estimated using data [46, 47, 48, 49, 50, 51, 52, 53]. Thus, we make the following assumption for robustification.

**Assumption 5.1** (Oracle for $\mathcal{H}_\infty$-Norm). An oracle to evaluate $\|G\|_\infty$ for any stable $G$ is accessible.

Now we are ready to present our robustification approach. For any $K$ satisfying $\rho(A - BK) < 1$, we define the following state-space representation:

$$\mathcal{T}(K) := \left[ \begin{array}{c|c} A - BK & C(R^w)^{-1/2} \\ \hline (Q + K^\top RK)^{1/2} & \mathbf{0} \end{array} \right]. \tag{5.1}$$

Properties of the $\mathcal{H}_\infty$-norm are well documented [54, 8, 41]. Now we state a few facts.

**Lemma 5.2.** The robust stability condition (Definition 3.3) on $K$ is equivalent to the frequency domain condition $\|\mathcal{T}(K)\|_\infty < 1$. Also, for any robustly stabilizing $K$ and $\widehat{K}$, there exists a continuous path connecting the two, such that every $K'$ on this path satisfies $\rho(A - BK') < 1$.

**Fact 5.3.** Denote the set of stabilizing $K$ as $\mathcal{K} := \{K \,|\, \rho(A - BK) < 1\}$. Then, $\|\mathcal{T}(K)\|_\infty$ is finite and is Clarke subdifferentiable at any $K \in \mathcal{K}$. Moreover, $\|\mathcal{T}(K)\|_\infty$ is an almost everywhere differentiable function of $K$.

The first half of Lemma 5.2 gives an alternative statement for the Bounded Real Lemma[2] [8, 41]. This connection makes the robust stability condition in Definition 3.3 more concrete and easier to verify, with a clearer robust control meaning: $\|\mathcal{T}(K)\|_\infty < 1$ means that the control gain $K$ *attenuates* the disturbance in the sense that the $\ell_2$-norm of the output $\{z_t\}_{t \geq 0}$ is smaller than the $R^w$-weighted $\ell_2$-norm of the disturbance $\{w_t\}_{t \geq 0}$. See more formal explanations in [7, 8]. The second half of Lemma 5.2 is a consequence of Lemma 4.6 in [55], and the fact that any robustly stabilizing $K$ must first be stabilizing.

Fact 5.3 restates the well-known fact [45, 56] that $\|\mathcal{T}(K)\|_\infty$ is possibly non-smooth but Clarke subdifferentiable [57]. These facts justify the applicability of a subgradient-descent-based optimization approach for decreasing the $\mathcal{H}_\infty$-norm, thus hopefully, can robustify the initialization to be $\|\mathcal{T}(K)\|_\infty < 1$. Specifically, if we have a stabilizing $K$ with a finite $\|\mathcal{T}(K)\|_\infty$ and $\|\mathcal{T}(K)\|_\infty > 1$, then the Clarke subgradient method can be used to decrease its $\mathcal{H}_\infty$-norm, until $\|\mathcal{T}(K)\|_\infty < 1$. It has been reported that this simple method is effective for minimizing $\mathcal{H}_\infty$-norms on most practical problems [58, 56]. Note that these Clarke subgradient methods require the model knowledge. When the model knowledge is unavailable, but only a (possibly noisy) $\mathcal{H}_\infty$-norm computation oracle exists, one can update $K$ following some derivative-free/zeroth-order approaches.

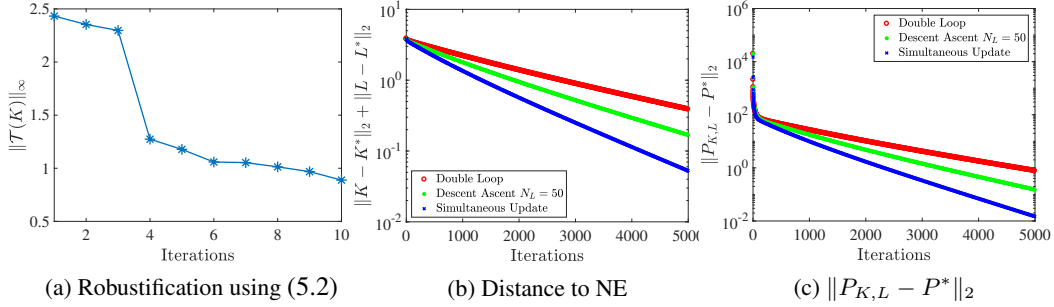

|  (a) Robustification using (5.2) | (b) Distance to NE | (c) $\|P_{K,L} - P^*\|_2$ |

Figure 2: Illustrating the effectiveness of the $\mathcal{H}_\infty$-based robustification using the two-point stochastic zeroth-order method, on the non-convergent case in §C.1.1. (a) shows the convergence of the update rule (5.2), when (C.2) is used for subgradient estimation. (b) and (c) show the convergence of all proposed RARL schemes after the robustification. The iterations denote the ones for updating $K$.

**Initialization Robustification.** Without model knowledge and under Assumption 5.1, one updates

$$K' = K - \alpha g, \tag{5.2}$$

where $g$ is some estimate for the subgradient of $\|\mathcal{T}(K)\|_\infty$ at $K$, and $\alpha > 0$ is the stepsize. One way is to estimate the $(i,j)$-th entry of the subgradient using the finite-difference method:

$$g_{ij} = \frac{\|\mathcal{T}(K + \epsilon d_{ij})\|_\infty - \|\mathcal{T}(K - \epsilon d_{ij})\|_\infty}{2\epsilon} \tag{5.3}$$

where $\epsilon$ is a small positive number and $d_{ij}$ is a matrix whose $(i,j)$-th entry is 1 and all other entries are 0. In general, to minimize $\|\mathcal{T}(K)\|_\infty$ over $K$, we need to consider the non-smoothness of $\|\mathcal{T}(K)\|_\infty$ and calculate the Clarke subgradients accurately. However, as $\|\mathcal{T}(K)\|_\infty$ is almost everywhere differentiable [45], the non-smoothness does not affect the optimization process in most iterates. We verify via simulations that this finite-difference method works well. We first show its effectiveness in the following example that has stability issues in §3.

**Example 5.4.** Consider the same one-dimensional system as in **Example** 3.2. As shown in Figure 1 (b), we first start with the same initialization as in **Example** 3.2, $(K_0, L_0) = (-4.6, 0)$, and then robustify $K_0$ using the update (5.2) with stepsize $\alpha = 0.007$, using finite-difference method (5.3) with $\epsilon = 10^{-8}$. The $\mathcal{H}_\infty$-norm indeed decreases. When $\|\mathcal{T}(K)\|_\infty < 1$, we use the updated $K$ as the robustified initialization. We then perform the descent-ascent update with $N_K = N_L = 1$, which is shown to converge to the NE very fast, with stepsize $\eta = 0.005$. This shows that our robustification technique indeed improves the original initialization, and guides the convergence of LQ RARL, even for the less stable case with $N_K = N_L = 1$.

When a noisy $\mathcal{H}_\infty$-norm oracle is used, a stochastic version of (5.3) can be calculated based on the *stochastic zeroth-order* methods [59, 60, 61]. Both one-point [59, 61] and two-point [62, 61] methods can be developed. See more details on the update rule and the simulation settings in C.4.2. As shown in Figure 2, the two-point approach efficiently robustifies the non-convergent case in §C.1.1 (developed for §3). With the robustified initialization, all three update rules converge successfully to the NE control gain $K^*$. More discussions and simulations on robustification are deferred to §C.4.

# 6 Concluding Remarks

In this paper, we have investigated the stability and convergence of policy-based robust adversarial RL, on the fundamental linear quadratic setup in continuous control. Several stability issues of LQ RARL have been identified, illustrating the intertwinement of both the initialization and update rule in developing provably convergent RARL algorithms. Through the lens of robust control, we have then proposed a provably stable and convergent initialization-update pair, and also developed $\mathcal{H}_\infty$-based approaches to robustify the initializations. Both our theoretical and empirical results have provided new and critical angles about RARL, from a rigorous robust control perspective. Interesting future directions include developing robustly stable RARL methods against some structured uncertainty, extending the robust control view to RARL in nonlinear systems, investigating the global convergence of descent-ascent methods, and studying the theoretical guarantees of our robustification approach.

## Broader Impact

We believe that researchers of reinforcement learning (RL), especially those who are interested in the theoretical foundations of *robust* RL, would benefit from this work, through the new insights and angles we have provided regarding robust adversarial RL (RARL) in linear quadratic (LQ) setups, from a rigorous *robust control* perspective. In particular, considering the impact of RARL [2] in RL with prominent empirical performance, and the ubiquity and fundamentality of LQ setups in continuous control, our results help pave the way for applying the RARL idea in control tasks.

More importantly, building upon the concepts from robust control, we have laid emphasis on the *robust stability* of RARL algorithms when applied to control systems, which has been overlooked in the RL literature, and is significant in continuous control, as a destabilized system can lead to catastrophic consequences. Such emphasis may encourage the development of more robust, and more importantly, *safe on-the-fly*, RARL algorithms, and push forward the development of RL for *safety-critical systems* as a whole. It also opens up the possibility to integrate more tools from the classic (robust) control theory, to improve the stability and robustness of popular RL algorithms practically used.

We do not believe that our research will cause any ethical issue, or put anyone at any disadvantage.

## Acknowledgments and Disclosure of Funding

The research of K.Z. and T.B. was supported in part by the US Army Research Laboratory (ARL) Cooperative Agreement W911NF-17-2-0196, and in part by the Office of Naval Research (ONR) MURI Grant N00014-16-1-2710.

## Footnotes

[1]Hereafter, we will use *policy* and *controller* interchangeably.

[2]There are a few equivalent forms of the Bounded Real Lemma, and this is another form with a frequency domain statement, as compared to Lemma 3.5.

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
