[Supplementary Material · supple.pdf]

# Supplementary Materials

## A Detailed Background

In this section, we provide a more detailed background on linear quadratic RARL and the corresponding zero-sum LQ games.

### A.1 LQ RARL and Robust Control

Consider a linear dynamical system

$$x_{t+1} = Ax_t + Bu_t + Cw_t, \tag{A.1}$$

where the system state is $x_t \in \mathbb{R}^d$, the control input of the agent at time $t$ is $u_t \in \mathbb{R}^{m_1}$, the disturbance or any unmodeled error at time $t$ is denoted by $w_t \in \mathbb{R}^{m_2}$. The matrices satisfy $A \in \mathbb{R}^{d \times d}$, $B \in \mathbb{R}^{d \times m_1}$, and $C \in \mathbb{R}^{d \times m_2}$. We define the one-stage cost as $c_t(x_t, u_t, w_t) := x_t^\top Q x_t + u_t^\top R^u u_t - w_t^\top R^w w_t$ with some positive definite matrices $Q \in \mathbb{R}^{d \times d}$, $R^u \in \mathbb{R}^{m_1 \times m_1}$, and $R^w \in \mathbb{R}^{m_2 \times m_2}$. Then the objective of the learning agent is to find $\{u_t\}_{t \geq 0}$ to minimize a cumulated cost subject to the worst-case disturbance:

$$\min_{\{u_t\}_{t\geq 0}} \sup_{\{w_t\}_{t\geq 0}} \mathcal{C}(\{u_t\}_{t\geq 0}, \{w_t\}_{t\geq 0}) := \mathbb{E}\left[\sum_{t=0}^{\infty} c_t(x_t, u_t, w_t)\right], \tag{A.2}$$

where the expectation is taken over the trajectory $\{x_t\}_{t\geq 0}$. For simplicity, we assume here that the only randomness stems from the initial state $x_0 \sim \mathcal{D}$, with some distribution $\mathcal{D}$ and $\mathbb{E}[x_0 x_0^\top] = \Sigma_0 > 0$.

In the zero-sum LQ dynamic game above, the disturbance is viewed as another player of the game besides the learning agent. The disturbance player, referred to as *adversary* hereafter, always plays against the agent by maximizing the long-term accumulated cost in (A.2). If the LQ game admits the *Nash equilibrium* (NE) control-disturbance sequences $\{u_t^*\}_{t\geq 0}$ and $\{w_t^*\}_{t\geq 0}$ such that the following inequality holds for any control-disturbance sequences $\{u_t\}_{t\geq 0}$ and $\{w_t\}_{t\geq 0}$,

$$\mathcal{C}(\{u_t^*\}_{t\geq 0}, \{w_t\}_{t\geq 0}) \leq \mathcal{C}(\{u_t^*\}_{t\geq 0}, \{w_t^*\}_{t\geq 0}) \leq \mathcal{C}(\{u_t\}_{t\geq 0}, \{w_t^*\}_{t\geq 0}), \tag{A.3}$$

then the value of the game exists and can be calculated as $\mathcal{C}(\{u_t^*\}_{t\geq 0}, \{w_t^*\}_{t\geq 0}) = \inf_{\{u_t\}_{t\geq 0}} \sup_{\{w_t\}_{t\geq 0}} \mathcal{C}(\{u_t\}_{t\geq 0}, \{w_t\}_{t\geq 0}) = \sup_{\{w_t\}_{t\geq 0}} \inf_{\{u_t\}_{t\geq 0}} \mathcal{C}(\{u_t\}_{t\geq 0}, \{w_t\}_{t\geq 0})$. Clearly, the NE control sequence $\{u_t^*\}_{t\geq 0}$ solves the original robust optimization problem (A.2). In the above setup, the "robust" control sequence coexists with the worst-case disturbance, i.e., the solution to the problem is a *pair* of sequences $(\{u_t^*\}_{t\geq 0}, \{w_t^*\}_{t\geq 0})$.

By (A.3), the NE control sequence $\{u_t^*\}_{t\geq 0}$ is *robust* to any disturbance, since even the worst-case disturbance $\{w_t^*\}_{t\geq 0}$ cannot drive the objective to infinity. Now we further explain the properties of $\{u_t^*\}_{t\geq 0}$ from a robust control perspective. Consider the minimization of $\mathbb{E}\sum_{t=0}^{\infty}(x_t^\top Q x_t + u_t^\top R^u u_t)$ subject to (2.1). If $\{w_t\}_{t\geq 0}$ is an i.i.d. Gaussian process, the problem becomes the state-feedback LQ Gaussian one, which shares the same solution as LQR. However, the statistical assumption on $\{w_t\}_{t\geq 0}$ can be unrealistic. The robust control framework removes this statistical assumption and considers the worst-case sequence from all $\{w_t\}_{t\geq 0}$ with bounded energy. Suppose the LQ game (A.2) admits the NE control-disturbance sequences $(\{u_t^*\}_{t\geq 0}, \{w_t^*\}_{t\geq 0})$. Let $\{x_t\}_{t\geq 0}$ be generated by the NE control sequence $\{u_t^*\}_{t\geq 0}$ and some disturbance sequence $\{w_t\}_{t\geq 0}$. We have the following inequality.

$$\sum_{t=0}^{\infty}(x_t^\top Q x_t + (u_t^*)^\top R^u u_t^*) \leq \mathcal{C}(\{u_t^*\}_{t\geq 0}, \{w_t^*\}_{t\geq 0}) + \sum_{t=0}^{\infty} ||w_t||_{R^w}^2$$

Therefore, when $\{u_t^*\}_{t\geq 0}$ is applied, any disturbance $\{w_t\}_{t\geq 0}$ can only further degrade the cost $\sum_{t=0}^{\infty}(x_t^\top Q x_t + (u_t^*)^\top R^u u_t^*)$ from the value $\mathcal{C}(\{u_t^*\}_{t\geq 0}, \{w_t^*\}_{t\geq 0})$ by an amount quantified by the weighted energy $\sum_{t=0}^{\infty} ||w_t||_{R^w}^2$. This provides some rationale for the game formulation of the robust control problem.

## A.2   Solution to LQ Dynamic Game

The solution to the above LQ game is characterized by the minimal positive semidefinite solution to the following generalized algebraic Riccati equation (GARE)

$$P^* = A^\top P^* A + Q - \begin{bmatrix} A^\top P^* B & A^\top P^* C \end{bmatrix} \begin{bmatrix} R^u + B^\top P^* B & B^\top P^* C \\ C^\top P^* B & -R^w + C^\top P^* C \end{bmatrix}^{-1} \begin{bmatrix} B^\top P^* A \\ C^\top P^* A \end{bmatrix}. \tag{A.4}$$

To ensure that the NE exists, we make the following standard assumption in robust control [63, 7, 64], which will be assumed throughout the paper.

**Assumption A.1.** There exists a minimal positive semidefinite solution $P^*$ to (A.4) that satisfies $R^w - C^\top P^* C > 0$.

The existence of the NE ensures the existence of the game value. More importantly, under Assumption A.1, the NE control-disturbance sequences can be generated by state-feedback in closed-forms. We rephrase Theorem 3.7 in [7] to validate this.

**Lemma A.2.** Under Assumption A.1, for any $x_0 \in \mathbb{R}^d$, as $Q > 0$, it follows that the LQ game admits the NE control-disturbance pair $(\{u_t^*\}_{t \geq 0}, \{w_t^*\}_{t \geq 0})$ generated by

$$u_t^* = -K^* x_t, \qquad w_t^* = -L^* x_t, \tag{A.5}$$

where $K^* \in \mathbb{R}^{m_1 \times d}$ and $L^* \in \mathbb{R}^{m_2 \times d}$ are given by

$$K^* = \{R^u + B^\top [P^* - P^* C(-R^w + C^\top P^* C)^{-1} C^\top P^*] B\}^{-1}$$
$$B^\top P^* [A - C(-R^w + C^\top P^* C)^{-1} C^\top P^* A], \tag{A.6}$$

$$L^* = \{-R^w + C^\top [P^* - P^* B(R^u + B^\top P^* B)^{-1} B^\top P^*] C\}^{-1}$$
$$C^\top P^* [A - B(R^u + B^\top P^* B)^{-1} B^\top P^* A]. \tag{A.7}$$

Moreover, we have $\mathcal{C}(\{u_t^*\}_{t \geq 0}, \{w_t^*\}_{t \geq 0}) = x_0^\top P^* x_0$, and the pair $(K^*, L^*)$ is stabilizing, i.e., $\rho(A - BK^* - CL^*) < 1$.

As per Lemma A.2, it suffices to find the pair of *stabilizing* control gain matrices $(K, L)$, such that (A.6)-(A.7) holds. One can apply policy-based RARL algorithms to directly search for such an NE in the matrix space $\mathbb{R}^{m_1 \times d} \times \mathbb{R}^{m_2 \times d}$.

## A.3   Policy-Based LQ RARL

By Lemma A.2, it suffices to find the NE $(K^*, L^*)$ within the stabilizing state feedback policies dictated by $u_t = -K x_t$, and $w_t = -L x_t$, with $(K, L) \in \mathbb{R}^{m_1 \times d} \times \mathbb{R}^{m_2 \times d}$ and $\rho(A - BK - CL) < 1$. This can be viewed as a basic benchmark for the policy-based RARL. Specifically, the protagonist aims at finding the robust control policy $K$, and the adversary tries to find the worst-case attack policy $L$. The objective of LQ RARL is to find $(K^*, L^*)$ using policy-based methods that solve the following minimax problem

$$\min_K \max_L \ \mathcal{C}(K, L) \tag{A.8}$$

such that for any stabilizing $K \in \mathbb{R}^{m_1 \times d}$ and $L \in \mathbb{R}^{m_2 \times d}$, $\mathcal{C}(K^*, L) \leq \mathcal{C}(K^*, L^*) \leq \mathcal{C}(K, L^*)$. The original RARL algorithm in [2] is applicable. Specifically, it alternates between the two players: the adversary improves its disturbing policy $L$ with the agent's policy $K$ fixed; the agent then learns its policy $K$ while holding the adversary's policy $L$ fixed. This sequence is repeated until convergence (if they do). We summarize such a scheme for LQ RARL as Algorithm 1. Note that we abstract out the update rule for $K_n$ and $L_n$ as *PolicyOptimizer* functions, which can be policy gradient, or natural PG updates that have been widely used in solving LQ problems [10, 15, 33]. We will present details for the *PolicyOptimizer* functions later in this section.

Here, we will reexamine the stability and convergence of the above algorithm and its variants on the proposed LQ RARL model. A few facts will be useful for our developments. First, the cost in (A.2) can be rewritten as

$$\mathcal{C}(K, L) := \mathbb{E}_{x_0 \sim \mathcal{D}} \left[ \sum_{t=0}^{\infty} x_t^\top (Q + K^\top R^u K - L^\top R^w L) x_t \right].$$

---

**Algorithm 1** Policy-Based LQ RARL Scheme [2]

---

**Input:** LQ RARL environment, initial policies $(K_0, L_0)$
**for** $n = 1, \ldots, N$ **do**
   Update $L_n \leftarrow L_{n-1}$
   **for** $j = 1, \ldots, N_L$ **do**
      Update $L_n \leftarrow$ PolicyOptimizer$(K_{n-1}, L_n)$
   **end for**
   Update $K_n \leftarrow K_{n-1}$
   **for** $i = 1, \ldots, N_K$ **do**
      Update $K_n \leftarrow$ PolicyOptimizer$(K_n, L_n)$
   **end for**
**end for**
**Return:** policy pair $(K_N, L_N)$

---

Let $P_{K,L}$ be the unique solution to the Lyapunov equation

$$P_{K,L} = Q + K^\top R^u K - L^\top R^w L + (A - BK - CL)^\top P_{K,L}(A - BK - CL). \tag{A.9}$$

Then for any *stablilizing* policy pair $(K, L)$, it follows that $\mathcal{C}(K, L) = \mathbb{E}_{x_0 \sim \mathcal{D}}\big(x_0^\top P_{K,L} x_0\big)$. In addition, we define $\Sigma_{K,L}$ as the state correlation matrix, i.e., $\Sigma_{K,L} := \mathbb{E}_{x_0 \sim \mathcal{D}} \sum_{t=0}^\infty x_t x_t^\top$. Then the policy gradient for the LQ RARL model can be calculated using the following lemma.

**Lemma A.3.** [33, 35] For discrete-time zero-sum LQ games, at any stabilizing policy pair $(K, L)$ with $\rho(A - BK - CL) < 1$, let $A_L := A - CL$ and $A_K := A - BK$, the policy gradients of $\mathcal{C}(K, L)$ have the following forms

$$\nabla_K \mathcal{C}(K, L) = 2[(R^u + B^\top P_{K,L} B)K - B^\top P_{K,L} A_L]\Sigma_{K,L}$$
$$\nabla_L \mathcal{C}(K, L) = 2[(-R^w + C^\top P_{K,L} C)L - C^\top P_{K,L} A_K]\Sigma_{K,L}.$$

Now we are ready to present details for the *PolicyOptimizer* functions. When $K$ is fixed, the *PolicyOptimizer* updates $L$ using either policy gradient or NPG:

$$L' = \begin{cases} L + \eta \nabla_L \mathcal{C}(K, L)\Sigma_{K,L}^{-1} & \text{if NPG} \\ L + \eta \nabla_L \mathcal{C}(K, L) & \text{if PG} \end{cases} ;$$

when $L$ is fixed, the *PolicyOptimizer* updates $K$ as:

$$K' = \begin{cases} K - \eta \nabla_K \mathcal{C}(K, L)\Sigma_{K,L}^{-1} & \text{if NPG} \\ K - \eta \nabla_K \mathcal{C}(K, L) & \text{if PG} \end{cases} ,$$

where $\eta > 0$ is some properly chosen stepsize. In practice, $\nabla_L \mathcal{C}(K, L)$, $\nabla_K \mathcal{C}(K, L)$, and $\Sigma_{K,L}$ can all be efficiently estimated from sample trajectories [10, 39]. As the focus of this paper is on the fundamental stability and optimization landscape, we assume exact updates in *PolicyOptimizer* are accessible. Note that by Lemma 3.2 in [33], the cost $\mathcal{C}(K, L)$ is nonconvex w.r.t. $K$ and nonconcave w.r.t. $L$, which makes finding the global Nash equilibrium NP-hard in general. However, we still have the following desired landscape of zero-sum LQ games, namely, that the stationary point implies the global NE under certain conditions. We summarize these existing results as follows for completeness.

**Lemma A.4** ([33], Lemma 3.2). There exists a $L \in \mathbb{R}^{m_2 \times d}$ such that $\min_K \mathcal{C}(K, L)$ is a nonconvex minimization problem; there exists a $K \in \mathbb{R}^{m_1 \times d}$, such that $\max_L \mathcal{C}(K, L)$ is a nonconcave maximization problem.

**Lemma A.5** ([33], Lemma 3.3). For a stabilizing policy pair $(K, L)$, i.e., $\rho(A - BK - CL) < 1$, suppose $\Sigma_{K,L}$ is full-rank and $(-R^w + C^\top P_{K,L} C)$ is invertible. If $\nabla_K \mathcal{C}(K, L) = \nabla_L \mathcal{C}(K, L) = 0$ and the induced $P_{K,L}$ defined in (2.5) is positive semidefinite, then $(K, L)$ constitutes the control gain pair at the Nash equilibrium.

## A.4 Understanding the Robust Stability Condition in Definition 3.3

Now we provide more explanations for understanding the robust stability condition in Definition 3.3, related to $\mathcal{H}_\infty$-control and the small gain theorem. From the Bounded Real Lemma (Lemma

3.5), the robust stability condition is equivalent to the feasibility of the LMI condition $M(P, K) < 0$ with some $P > 0$. Based on this LMI condition, one can apply either $S$-procedure or the dissipation inequality argument to show that the dynamics $x_{t+1} = Ax_t + Bu_t + Cw_t$ is robustly stable for all $\{w_t\}_{t\geq 0}$ satisfying a small gain condition, i.e., $\sum_{t=0}^{N} w_t^\top R^w w_t \leq \sum_{t=0}^{N} x_t^\top (Q + K^\top RK) x_t$ for all $N$. This can be viewed as a variant of the well-known small gain theorem [65]. Notice that the above small gain condition on $\{w_t\}_{t\geq 0}$ is general enough to cover many types of uncertainty models used in robust control. For illustrative purposes, we review a few well-known examples here.

- Parametric uncertainty: Consider a linear system $x_{t+1} = (\widehat{A} - BK)x_t$ where $K$ is designed to control the system. Typically one does not know $\widehat{A}$ exactly. In contrast, one has $\widehat{A} = A + A_\delta$ where $A$ is some estimation of $\widehat{A}$ and $A_\delta$ captures the uncertainty in the system dynamics. Although the exact value of $A_\delta$ is not known, the norm of this matrix is small and can be bounded above. Therefore, the system dynamics becomes $x_{t+1} = (A - BK)x_t + w_t$ with $w_t = A_\delta x_t$. In this case, one can choose $R^w$ properly so that the norm bound for $A_\delta$ is transformed into the small gain condition on $\{w_t\}_{t\geq 0}$.

- Time-varying parameters: In the above example, we can further allow $A_\delta$ to change over time, i.e. $w_t = (A_\delta^{(t)})x_t$. As long as there is a uniform upper bound on the norm of $A_\delta^{(t)}$, we will be able to obtain a small gain condition on $\{w_t\}_{t\geq 0}$.

- Dynamical uncertainty: Sometimes even the order of the model may be uncertain. For example, one may use a rigid body model for control purposes when there are flexible modes in the true dynamics. In this case, $w_t$ is not completely determined by $x_t$. Importantly, the computation of $w_t$ may require that the past information of the sequence $\{x_t\}_{t\geq 0}$, and $\{w_t\}_{t\geq 0}$ may be generated as the output of some dynamical system whose state dimension is even unknown. However, one may still be able to obtain some upper bound on the $\mathcal{H}_\infty$-norm of this unknown system and use this information to get the small gain condition on $\{w_t\}_{t\geq 0}$.

- Nonlinearity: Sometimes the true system is nonlinear and governed by the model $x_{t+1} = (A - BK)x_t + \phi(x_t)$ where $\phi$ is some nonlinear function. Then one can model $w_t$ as $w_t = \phi(x_t)$ and use the property of $\phi$ to construct the small gain condition.

From the above discussion, one can see that the robust stability condition in Definition 3.3 can ensure that the controller $K$ robustly stabilizes the system, even in the presence of "small" model uncertainty and system nonlinearity. From a game-theoretic perspective, the robust stability condition also ensures that the inner-loop problem $\max_L \mathcal{C}(K, L)$ is well-defined, see Lemma 3.4, which is substantiated as follows.

### A.4.1 Proof of Lemma 3.4

The proof follows by applying Theorem 3.7 in [7] on the new system with the matrices $A$, $B$, $Q$, $\gamma^2 I$ in the book being replaced by $A - BK$, $0$, $Q + K^\top R^u K$, $R^w$. As $Q > 0$, the pair $(A - BK, Q + K^\top R^u K)$ is detectable. Thus, for this fixed $K$, the GARE (3.52b) in [7] reduces to the ARE (3.1), and the condition (3.53) therein is equivalent to our $R^w - C^\top P_K^* C > 0$ condition in Definition 3.3, for the minimal positive definite solution $P_K^*$ (note that by (ii) of Theorem 3.7 in [7], existence of a positive semidefinite solution satisfying this condition is *equivalent* to the existence of a minimal one that satisfies this). Thus, by applying Theorem 3.7 (ii), the inner-loop problem admits a *finite* value, given by $\mathbb{E}_{x_0 \sim \mathcal{D}}(x_0^\top P_K^* x)$. Applying Theorem 3.7 (iv) gives the form of the maximization feedback solution (Eq. (3.51b) therein), as given in the lemma. By the theory of ARE [66, 63], we know that this $L(K)$ is stabilizing, and is unique among all stabilizing disturbance gain matrices $L$ for the system $(A - BK, C)$, i.e., those that make $\rho(A - BK - CL) < 1$. Indeed, for any stabilizing $L$, by comparing the Lyapunov equation (A.9) with the Riccati equation (3.1), letting $A_{K,L} = A - BK - CL$, we have

$$P_K^* - P_{K,L} = A_{K,L}^\top (P_K^* - P_{K,L}) A_{K,L} - (L(K) - L)^\top (-R^w + C^\top P_K^* C)(L(K) - L),$$

which, by $R^w - C^\top P_K^* C > 0$, implies $P_K^* \geq P_{K,L}$. This further means that for all stabilizing $L$,

$$R^w - C^\top P_{K,L} C > 0. \tag{A.10}$$

On the other hand, for fixed $K$ and $\Sigma_0 > 0$ (thus $\Sigma_{K,L} > 0$), by setting the gradient $\nabla_L \mathcal{C}(K, L) = 0$ and Lemma A.3, we know that all the stationary points (if there are multiple) $\check{L}$ satisfy

$$(-R^w + C^\top P_{K,\check{L}} C)L - C^\top P_{K,\check{L}} A_K = 0. \tag{A.11}$$

Moreover, suppose that there are two such stabilizing stationary-points $\check{L}_1, \check{L}_2$. By comparing (A.9) associated with $\check{L}_1, \check{L}_2$, we have

$$P_{K,\check{L}_1} - P_{K,\check{L}_2} = A_{K,\check{L}_2}^\top (P_{K,\check{L}_1} - P_{K,\check{L}_2})A_{K,\check{L}_2} - (\check{L}_1 - \check{L}_2)^\top(-R^w + C^\top P_{K,\check{L}_1}C)(\check{L}_1 - \check{L}_2),$$

$$P_{K,\check{L}_2} - P_{K,\check{L}_1} = A_{K,\check{L}_1}^\top (P_{K,\check{L}_2} - P_{K,\check{L}_1})A_{K,\check{L}_1} - (\check{L}_2 - \check{L}_1)^\top(-R^w + C^\top P_{K,\check{L}_2}C)(\check{L}_2 - \check{L}_1),$$

where (A.11) has been substituted in. This further implies $P_{K,\check{L}_1} \geq P_{K,\check{L}_2}$ and $P_{K,\check{L}_1} \leq P_{K,\check{L}_2}$, respectively, since $R^w - C^\top P_{K,\check{L}_i}C > 0$ for both $i = 1, 2$ (by (A.10)). This shows that the stationary point is in fact unique. This unique stationary point thus has to be that $L(K)$ given before, which completes the proof. $\qquad\square$

# B  Proofs

In this section, we provide proofs for the theoretical results in the main paper.

## B.1  Proof of Lemma 4.1

Our proof is inspired by the matrix inequality technique developed in [34]. By Lemma 3.5, $K_{n+1}$ satisfies the robust stability condition if there exists $P > 0$ such that $M(P, K_{n+1}) < 0$. Obviously, the Riccati equation (3.1) can be used to show $P_{K_n}^* \geq 0$ and $M(P_{K_n}^*, K_n) \leq 0$. Due to the NPG update (4.1), we can further verify $M(P_{K_n}^*, K_{n+1}) \leq 0$ using the Schur complement lemma. Next, notice $P_{K_n}^*$ can be further perturbed into a solution for the strict matrix inequality condition. Since $\rho(A - BK_n - CL(K_n)) < 1$, we can obtain a positive definite matrix $\bar{P}_{K_n}$ satisfying the Lyapunov equation $(A - BK_n - CL(K_n))^\top \bar{P}_{K_n}(A - BK_n - CL(K_n)) - \bar{P}_{K_n} = -I$. We can show $M(P_{K_n}^* + \alpha\bar{P}_{K_n}, K_{n+1}) < 0$ for sufficiently small $\alpha > 0$. Therefore, $K_{n+1}$ satisfies the robust stability condition. More details are provided as follows.

We now need to show that there exists a positive definite matrix $P$ such that $M(P, K_{n+1}) < 0$. Our proof includes two steps. In the first step, we prove the non-strict matrix inequality $M(P_{K_n}^*, K_{n+1}) \leq 0$. In the second step, we perturb $P_{K_n}^*$ such that the strict matrix inequality condition holds.

**Step 1: Feasibility of the non-strict matrix inequality.**   In this step, we verify $M(P_{K_n}^*, K_{n+1}) \leq 0$. By Schur complement lemma, we only need to verify $C^\top P_{K_n}^* C - R^w \leq 0$ (which is obviously true due to the robust stability condition on $K_n$) and another matrix inequality $M_1(P_{K_n}^*, K_{n+1}) \leq 0$ where $M_1$ maps any pair $(P, K)$ to another matrix as

$$M_1(P, K) = (A - BK)^\top P(A - BK) - P + Q + K^\top R^u K$$
$$- (A - BK)^\top PC(C^\top PC - R^w)^{-1}C^\top P(A - BK).$$

For clarification, we note that $M_1(P_{K_n}^*, K_{n+1}) \leq 0$ is equivalent to the following matrix inequality:

$$(A - BK_{n+1})^\top P_{K_n}^*(A - BK_{n+1}) - P_{K_n}^* + Q + K_{n+1}^\top R^u K_{n+1}$$
$$\leq (A - BK_{n+1})^\top P_{K_n}^* C(C^\top P_{K_n}^* C - R^w)^{-1}C^\top P_{K_n}^*(A - BK_{n+1}).$$

As discussed in the main paper, $P_{K_n}^*$ is the unique stabilizing positive semidefinite solution to the Riccati equation $M_1(P_{K_n}^*, K_n) = 0$. Recall that $K_{n+1}$ is updated using the NPG rule

$$K_{n+1} = K_n - \eta\nabla_K \mathcal{C}(K_n, L(K_n))\Sigma_{K_n, L(K_n)}^{-1}$$
$$= K_n - 2\eta((R^u + B^\top P_{K_n}^* B)K_n - B^\top P_{K_n}^*(A - CL(K_n)))$$
$$= K_n - 2\eta((R^u + B^\top \widetilde{P}_{K_n}^* B)K_n - B^\top \widetilde{P}_{K_n}^* A),$$

where $L(K_n) = (-R^w + C^\top P_{K_n}^* C)^{-1}C^\top P_{K_n}^*(A - BK_n)$ and $\widetilde{P}_{K_n}^* = P_{K_n}^* + P_{K_n}^* C(C^\top P_{K_n}^* C - R^w)^{-1}C^\top P_{K_n}^*$. Therefore, a straightforward calculation yields

$$M_1(P_{K_n}^*, K_{n+1}) - M_1(P_{K_n}^*, K_n)$$
$$= (K_n - K_{n+1})^\top B^\top \widetilde{P}_{K_n}^*(A - BK_n) + (A - BK_n)^\top \widetilde{P}_{K_n}^* B(K_n - K_{n+1})$$
$$+ (K_n - K_{n+1})^\top(B^\top \widetilde{P}_{K_n}^* B + R^u)(K_n - K_{n+1}) + (K_{n+1} - K_n)^\top R^u K_n$$
$$+ K_n^\top R^u(K_{n+1} - K_n).$$

Notice the following relation:

$$(K_n - K_{n+1})^\top B^\top \widetilde{P}^*_{K_n}(A - BK_n) + (K_{n+1} - K_n)^\top R^u K_n$$
$$= (K_{n+1} - K_n)^\top ((B^\top \widetilde{P}^*_{K_n} B + R^u)K_n - B^\top \widetilde{P}^*_{K_n} A).$$

Therefore, we can show

$$M_1(P^*_{K_n}, K_{n+1}) - M_1(P^*_{K_n}, K_n) = E^\top_{K_n, L(K_n)} \left(-4\eta I + 4\eta^2 (B^\top \widetilde{P}^*_{K_n} B + R^u)\right) E_{K_n, L(K_n)}$$

where

$$E_{K_n, L(K_n)} = (R^u + B^\top P^*_{K_n} B)K_n - B^\top P^*_{K_n}(A - CL(K_n))$$
$$= (R^u + B^\top \widetilde{P}^*_{K_n} B)K_n - B^\top \widetilde{P}^*_{K_n} A.$$

Based on the assumption on $\eta$, we know $-4\eta I + 4\eta^2 (B^\top \widetilde{P}^*_{K_n} B + R^u) \leq 0$. Therefore, we have

$$M_1(P^*_{K_n}, K_{n+1}) = M_1(P^*_{K_n}, K_n) + E^\top_{K_n, L(K_n)} \left(-4\eta I + 4\eta^2 (B^\top \widetilde{P}^*_{K_n} B + R^u)\right) E_{K_n, L(K_n)}$$
$$\leq M_1(P^*_{K_n}, K_n) = 0.$$

**Step 2: Feasibility of the strict matrix inequality via perturbation.** As $\rho(A - BK_n - CL(K_n)) < 1$, we know that there exists a unique positive definite matrix $\bar{P}_{K_n}$ such that the following equation holds

$$(A - BK_n - CL(K_n))^\top \bar{P}_{K_n}(A - BK_n - CL(K_n)) - \bar{P}_{K_n} = -I.$$

If we choose a sufficiently small positive scalar $\epsilon > 0$, then we will have $M(P^*_{K_n} + \epsilon \bar{P}_{K_n}, K_{n+1}) < 0$. In other words, we can perturb the non-strict matrix inequality solution $P^*_{K_n}$ into a strict matrix inequality solution $P^*_{K_n} + \epsilon \bar{P}_{K_n}$. To prove this, we only need to verify $C^\top (P^*_{K_n} + \epsilon \bar{P}_{K_n})C - R^w < 0$ and $M_1(P^*_{K_n} + \epsilon \bar{P}_{K_n}, K_{n+1}) < 0$. We know $C^\top P^*_{K_n} C - R^w < 0$ and hence there exists a sufficiently small $\epsilon$ such that $C^\top (P^*_{K_n} + \epsilon \bar{P}_{K_n})C - R^w < 0$. Next, after some lengthy (but tedious) calculations, we can use the matrix inversion lemma to show

$$M_1(P^*_{K_n} + \epsilon \bar{P}_{K,n}, K_{n+1})$$
$$= M_1(P^*_{K_n}, K_{n+1}) + \epsilon \left((A - BK_n - CL(K_n))^\top \bar{P}_{K_n}(A - BK_n - CL(K_n)) - \bar{P}_{K_n}\right) + o(\epsilon)$$
$$= M_1(P^*_{K_n}, K_{n+1}) + \epsilon I + o(\epsilon^2).$$

Therefore, $M_1(P^*_{K_n} + \epsilon \bar{P}_{K,n}, K_{n+1}) > 0$ for a sufficiently small $\epsilon$. This completes the proof. $\qquad\square$

**Remark B.1** (Comparison to Existing Results). Notice that a similar result is also established in Section 9 (Lemma 9.2) of [35]. However, the proof techniques therein is based on a contradiction argument, and is different from our linear matrix inequality-based techniques, which we first developed in [34]. In addition, the assumption (Assumption 1) in [35] is intended for general zero-sum LQ games, while our assumption (Assumption A.1) follows the standard $\mathcal{H}_\infty$-control literature. The robust stability condition adopted in our theory has additional robust control implications and has been used to develop our proposed $\mathcal{H}_\infty$-based initialization technique. Note that both results improve the previous results in [33], where the equivalent notion of "robust stability" was enforced by a projection step, which can be restrictive and also has few robust control implications.

## B.2 Proof of Theorem 4.2

With Lemma 4.1 being proved, this theorem can now be proved using the cost difference lemma (see Lemma 6.7 in [33] and Lemma 5.1 in [34]) and a standard sublinear rate argument routine (e.g. see Section 6.3 in [33] and Section 5.2 in [34]). Specifically, we can merge the Riccati equations $M_1(P^*_{K_{n+1}}, K_{n+1}) = 0$ and $M_1(P^*_{K_n}, K_n) = 0$ to obtain the one-step descent $\text{Tr}(P^*_{K_{n+1}}) - \text{Tr}(P^*_{K_n}) \leq -2\eta \text{Tr}\left(E^\top_{K_n, L(K_n)} E_{K_n, L(K_n)}\right)$. In fact, following Section 5.2 in [34], a matrix-wise descent is guaranteed, i.e., $P^*_{K_{n+1}} \leq P^*_{K_n}$. This implies a uniform stepsize choice of $\eta \leq$

$1/(2\|R^u + B^\top \widetilde{P}^*_{K_0} B\|)$. Therefore, we can sum up the inequality from $n = 0$ to some $N > 0$, and obtain the following result

$$\frac{1}{N+1}\sum_{n=0}^{N}\mathrm{Tr}\left(E^\top_{K_n,L(K_n)}E_{K_n,L(K_n)}\right) \le \frac{\mathrm{Tr}(P^*_{K_0})}{2\eta(N+1)},$$

which demonstrates that the outer-loop iteration $K_n$ converges to the stationary point $K^*$ satisfying $E_{K^*,L(K^*)} = 0$ with the $O(1/N)$ rate as promised. Note that by Lemma A.5, this stationary point is the unique NE solution $K^*$, which completes the proof. $\qquad\square$

### B.3 Proof of Lemma 4.4

We first prove the following supporting lemma.

**Lemma B.2.** Given a fixed policy $K$, let $A_K = A - BK$. Suppose $K$ satisfies the robust stability condition (thus $R^w - C^\top P^*_K C > 0$). For any $L$ satisfying $\rho(A_K - CL) < 1$, $P_{K,L} \le P^*_K$, and $R^w - C^\top P_{K,L}C > 0$, if $\eta_L \le 1/(2\|R^w - C^\top P_{K,L}C\|)$, then

$$\mathcal{C}(K, L(K)) - \mathcal{C}\left(K, L + \eta_L\nabla_L\mathcal{C}(K,L)\Sigma^{-1}_{K,L}\right) \le (1 - C_0\eta_L)(\mathcal{C}(K,L(K)) - \mathcal{C}(K,L)), \quad \text{(B.1)}$$

where $C_0 = 2\mu\|\Sigma_{K,L(K)}\|^{-1}\lambda_{\min}\left(R^w - C^\top P^*_K C\right)$ with $\mu := \lambda_{\min}(\Sigma_0) > 0$.

*Proof.* Since $K$ is fixed, the dynamical model becomes $x_{t+1} = A_K x_t + Cw_t$. The one-stage cost is $x_t^\top(Q + K^\top R^u K)x_t + w_t^\top(-R^w)w_t$. Since $(Q + K^\top R^u K)$ is positive definite and $(-R^w)$ is negative definite, the inner-loop optimization is a nonstandard LQR problem, with a negative definite cost weighting matrix $-(Q + K^\top R^u K)$. The cost difference formula for the standard LQR still applies [10, 33, 35, 67]. Specifically, if $(K, L)$ and $(K, L')$ are both stabilizing, one can verify that

$$P_{K,L'} - P_{K,L} \qquad\qquad\qquad\qquad\qquad\qquad\qquad\qquad\qquad\qquad\qquad\qquad\qquad \text{(B.2)}$$

$$= \sum_{k=0}^{\infty}((A_K - CL')^\top)^k\left(\Delta_L^\top E_{K,L} + E_{K,L}^\top\Delta_L + \Delta_L^\top(C^\top P_{K,L}C - R^w)\Delta_L\right)(A_K - CL')^k,$$

where $E_{K,L} = (-R^w + C^\top P_{K,L}C)L - C^\top P_{K,L}A_K$ and $\Delta_L = L' - L$. In the above formula, the only requirement on $L'$ is $\rho(A_K - CL') < 1$.

**Step 1: Stability of the NPG update.** We first use (B.2) to show $(K, L + \eta_L\nabla_L\mathcal{C}(K,L)\Sigma^{-1}_{K,L})$ is stabilizing given $\eta_L \le 1/(2\|R^w - C^\top P_{K,L}C\|)$. Since $\rho(A_K - CL) < 1$ and the spectral radius is continuous, there exists $\zeta$ such that $\rho\left(A_K - C(L + \eta_L\nabla_L\mathcal{C}(K,L)\Sigma^{-1}_{K,L})\right) < 1$ for any $\eta_L < \zeta$ and $\rho\left(A_K - C(L + \zeta\nabla_L\mathcal{C}(K,L)\Sigma^{-1}_{K,L})\right) = 1$. Now we use contradiction to show $\zeta > 1/(2\|R^w - C^\top P_{K,L}C\|)$. Suppose $\zeta \le 1/(2\|R^w - C^\top P_{K,L}C\|)$. Consider $L' = L + \eta_L\nabla_L\mathcal{C}(K,L)\Sigma^{-1}_{K,L} = L + 2\eta_L E_{K,L}$ where $\eta_L < \zeta$. Clearly, $(K, L')$ is a stabilizing pair by assumption. We have $\Delta_L = 2\eta_L E_{K,L}$. Hence, (B.2) yields

$$P_{K,L'} - P_{K,L}$$

$$= \sum_{k=0}^{\infty}((A_K - CL')^\top)^k\left(4\eta_L E_{K,L}^\top E_{K,L} + 4\eta_L^2 E_{K,L}^\top(C^\top P_{K,L}C - R^w)E_{K,L}\right)(A_K - CL')^k$$

$$\ge \sum_{k=0}^{\infty}((A_K - CL')^\top)^k\left(2\eta_L E_{K,L}^\top E_{K,L}\right)(A_K - CL')^k.$$

The last step relies on the assumption that $\eta_L < \zeta \le 1/(2\|R^w - C^\top P_{K,L}C\|)$. Therefore, $P_{K,L'} \ge P_{K,L}$ for the NPG update $L' = L + \eta_L\nabla_L\mathcal{C}(K,L)\Sigma^{-1}_{K,L}$ with $\eta_L < \zeta$.

Next, notice that $\rho(A_K - CL(K)) < 1$ and $E_{K,L(K)} = 0$. Then (B.2) can be used to show the following equation for any $L' = L + \eta_L\nabla_L\mathcal{C}(K,L)\Sigma^{-1}_{K,L}$ with $\eta_L < \zeta$:

$$P_{K,L'} - P_{K,L(K)} = (A_K - CL')^\top(P_{K,L'} - P_{K,L(K)})(A_K - CL')$$
$$+ (L' - L(K))(C^\top P_{K,L(K)}C - R^w)(L' - L(K)). \qquad \text{(B.3)}$$

Notice $R^w - C^\top P_{K,L(K)}C > 0$. Hence we have $P_{K,L} \leq P_{K,L'} \leq P_{K,L(K)}$. Now we consider a sequence $\zeta_l \in [0, \zeta)$ such that $\lim_{l \to \infty} \zeta_l = \zeta$. We denote $L_{\zeta_l} = L + \zeta_l \nabla_L \mathcal{C}(K, L) \Sigma_{K,L}^{-1}$. Clearly $P_{K,L} \leq P_{K,L_{\zeta_l}} \leq P_{K,L(K)}$ for all $\zeta_l$. The bounded sequence $\{P_{K,L_{\zeta_l}}\}_{l \geq 0}$ has at least one limit point, and we denote this limit point as $Z$. Now we take the subsequence of $\{P_{K,L_{\zeta_l}}\}_{l \geq 0}$ which converges to $Z$ and take the limit of this subsequence on both sides of (B.3). This leads to the following equation

$$
\begin{aligned}
Z - P_{K,L(K)} = {} & (A_K - CL_\zeta)^\top (Z - P_{K,L(K)})(A_K - CL_\zeta) \\
& + (L_\zeta - L(K))(C^\top P_{K,L(K)}C - R^w)(L_\zeta - L(K)). \quad \text{(B.4)}
\end{aligned}
$$

By assumption $\rho(A_K - CL_\zeta) = 1$, there exists a vector $v$ satisfying $(A_K - CL_\zeta)v = \lambda v$ and $|\lambda| = 1$. Combining the facts $|\lambda| = 1$, $R^w - C^\top P_{K,L(K)}C > 0$, with (B.4), we can show $L_\zeta v = L(K)v$. Therefore, $(A_K - CL(K))v = \lambda v$ with $|\lambda| = 1$. However, we already know $\rho(A_K - CL(K)) < 1$. This is a contradiction. Therefore, it is impossible to have $\zeta \leq 1/(2\|R^w - C^\top P_{K,L}C\|)$ in the first place. Consequently, we have $\zeta > 1/(2\|R^w - C^\top P_{K,L}C\|)$, and $(K, L + \eta_L \nabla_L \mathcal{C}(K, L)\Sigma_{K,L}^{-1})$ is stabilizing given $\eta_L \leq 1/(2\|R^w - C^\top P_{K,L}C\|)$.

**Step 2: Convergence bound.** We have shown $\rho\left(A_K - C(L + \eta_L \nabla_L \mathcal{C}(K, L)\Sigma_{K,L}^{-1})\right) < 1$. An immediate consequence is that $\mathcal{C}\left(K, L + \eta_L \nabla_L \mathcal{C}(K, L)\Sigma_{K,L}^{-1}\right)$ is guaranteed to be finite. The rest of the proof is standard. Recall that $\mu = \lambda_{\min}\left(\mathbb{E}_{x_0 \sim \mathcal{D}}[x_0 x_0^\top]\right)$. By applying (B.2), we can obtain the so-called *almost smoothness condition* [10]

$$
\mathcal{C}(K, L') - \mathcal{C}(K, L) = 2\,\mathrm{Tr}\left(\Sigma_{K,L'}\Delta_L^\top E_{K,L}\right) + \mathrm{Tr}\left(\Sigma_{K,L'}\Delta_L^\top (C^\top P_{K,L}C - R^w)\Delta_L\right). \quad \text{(B.5)}
$$

Applying completion of squares, we can obtain

$$
\begin{aligned}
\mathcal{C}(K, L(K)) - \mathcal{C}(K, L) &= 2\,\mathrm{Tr}\left(\Sigma_{K,L(K)}\Delta_L^\top E_{K,L}\right) + \mathrm{Tr}\left(\Sigma_{K,L(K)}\Delta_L^\top (C^\top P_{K,L}C - R^w)\Delta_L\right) \\
&\leq \mathrm{Tr}\left(\Sigma_{K,L(K)}E_{K,L}^\top (R^w - C^\top P_{K,L}C)^{-1}E_{K,L}\right) \\
&\leq \|\Sigma_{K,L(K)}\|\lambda_{\min}\left(R^w - C^\top P_{K,L(K)}C\right)^{-1}\mathrm{Tr}(E_{K,L}^\top E_{K,L}),
\end{aligned}
$$
(B.6)

where the last step relies on the fact $P_{K,L} \leq P_{K,L(K)}$. Now we can set $L' = L + 2\eta_L E_{K,L}$ in (B.5) to show

$$
\begin{aligned}
\mathcal{C}\left(K, L + \eta_L \nabla_L \mathcal{C}(K, L)\Sigma_{K,L}^{-1}\right) - \mathcal{C}(K, L) &\geq 2\eta_L \mu\,\mathrm{Tr}(E_{K,L}^\top E_{K,L}) \\
&\geq \frac{2\eta_L \mu \lambda_{\min}(R^w - C^\top P_{K,L(K)}C)}{\|\Sigma_{K,L(K)}\|}(\mathcal{C}(K, L(K)) - \mathcal{C}(K, L))
\end{aligned}
$$

which directly leads to the desired conclusion. $\qquad\square$

Now we return to prove Lemma 4.4. It remains to use induction to show that $P_{K,L}$ is nondecreasing and the condition $\eta_L \leq 1/(2\|R^w - C^\top P_{K,L}C\|)$ is guaranteed (as we update $L$), due to the assumption that $\eta_L \leq 1/(2\|R^w - C^\top P_{K,L_0}C\|)$ for the stabilizing initialization $L_0$. Therefore we can apply Lemma B.2 at every step in the inner-loop optimization and obtain a convergence bound with a linear rate $(1 - C_0 \eta_L)^k$ with $C_0 = 2\mu\|\Sigma_{K,L(K)}\|^{-1}\lambda_{\min}\left(R^w - C^\top P_{K,L(K)}C\right)$. $\qquad\square$

### B.4 Discussions on Local Results

Much faster rates, i.e., (super-)linear rates, can be shown locally around $K^*$. We summarize the results as follows. As they are not the focus of our paper, we only provide proof sketches/ideas.

**Theorem B.3.** Suppose $K_0$ satisfies the robust stability condition.

1. Consider the NPG update with $\eta \leq 1/(2\|R^u + B^\top \widetilde{P}_{K_0}^* B\|)$. The outer-loop iteration achieves local linear rate.

2. Suppose we update the outer-loop iteration using the Gauss-Newton method with stepsize $\eta = 1/2$, i.e.,

$$K_{n+1} = (R^u + B^\top \widetilde{P}^*_{K_n} B)^{-1} B^\top \widetilde{P}^*_{K_n} A.$$

Then this method achieves locally Q-quadratic rates.

*Proof Sketch.* The proof can be established based on the standard argument which expands the Taylor series of $\widetilde{P}^*_{K_n}$ around $K_n$. For NPG, the resultant system is linear. For Gauss-Newton method, the resultant system is quadratic. Explicit linear rates for NPG can also be obtained by the arguments in [34, 33], e.g., see the proof of Theorem 4.6 in [34] and Theorem 5.3 in [33]. □

We note that the Gauss-Newton method can hardly be implemented in a model-free manner using derivative-free methods [10, 34, 35], and thus is not the focus of our main paper. Finally, we comment that similar linearization arguments can be combined with the *Lyapunov's indirect method* [65], to show the local linear convergence of the descent-ascent algorithm. The Taylor expansion calculations are straightforward and we do not include the details here.

## C   Simulations

In this section, we provide more detailed simulation results to support our findings in the main paper.

### C.1   Results for Section §3

We now show that the two stability issues identified in §3 can also easily occur in cases beyond the one-dimension ones.

#### C.1.1   Stability issue due to bad initialization

Consider the system where

$$A = \begin{bmatrix} 1.2763 & 0.9780 & 0.2684 & 0.3809 \\ 0.7799 & 1.7385 & 0.4999 & 0.0659 \\ 0.4384 & 0.5011 & 1.8792 & 0.2881 \\ 0.7235 & 0.0721 & 0.8037 & 2.1096 \end{bmatrix}, \quad B = \begin{bmatrix} 0.2134 & 0.6005 \\ 0.4521 & 0.9501 \\ 0.9312 & 0.2303 \\ 0.0249 & 0.5485 \end{bmatrix}, \quad C = \begin{bmatrix} 0.0182 & 0.0134 \\ 0.0027 & 0.0094 \\ 0.0105 & 0.0041 \\ 0.0150 & 0.0098 \end{bmatrix},$$

$Q = 1.0477 \cdot I, \quad R^u = 1.3724 \cdot I, \quad R^w = 1036.2265 \cdot I.$

We select the initialization $K_0$ to be

$$K_0 = \begin{bmatrix} -0.6342 & 0.0619 & 1.3209 & -1.1933 \\ 1.5035 & 0.5671 & 0.8931 & 2.3151 \end{bmatrix}.$$

One can easily check that $K_0$ is stabilizing, i.e., $\rho(A - BK) = 0.9958 < 1$. However, the $\mathcal{H}_\infty$-norm $\|\mathcal{T}(K_0)\|_\infty = 2.4325 > 1$, i.e., $K_0$ is not robustly stabilizing (see Lemma 5.2).

Now we adopt the conventional RARL scheme in [2], where we first fix $K_0$, and try to optimize $L$ to obtain $L(K_0)$ using the NPG update in §2.2. The stepsize $\eta_L$ for $L$ update is chosen as infinitesimal as $\eta_L = 10^{-10}$. As shown in Figure 3, both the spectral radius $\rho(A - BK_0 - CL)$ and the distance to the NE $\|K - K^*\|_2 + \|L - L^*\|_2$ blow up, even with such an infinitesimal stepsize. Particularly, the updated $L$ destabilizes the system under the control $K_0$. The essential reason behind this non-convergence result is that for a fixed $K_0$ that is not robustly stabilizing, $\max_L \mathcal{C}(K_0, L)$ does not yield a solution. Indeed, `Matlab` returns a solvability issue for solving the inner-loop Riccati equation (3.1) induced by $K_0$. This shows that even a stabilizing initialization $K_0$ is not enough for LQ RARL, as the solution $L(K_0)$ may either not necessarily exist, or destabilize the system, similar to what we have observed in **Example** 3.2.

#### C.1.2   Stability issue due to bad choices of $(N_K, N_L)$

Consider the system where

$$A = \begin{bmatrix} 1.7865 & 0.3912 & 0.8758 & 0.5996 \\ 0.2756 & 1.3175 & 0.7692 & 0.4848 \\ 0.4764 & 0.9786 & 1.0618 & 0.7591 \\ 0.4489 & 0.7918 & 0.6014 & 1.7520 \end{bmatrix}, \quad B = \begin{bmatrix} 0.1303 & 0.0312 \\ 0.1309 & 0.0528 \\ 0.7452 & 0.6727 \\ 0.2460 & 0.0743 \end{bmatrix}, \quad C = \begin{bmatrix} 0.0058 & 0.0028 \\ 0.0015 & 0.0116 \\ 0.0188 & 0.0169 \\ 0.0056 & 0.0026 \end{bmatrix},$$

$Q = 1.0613 \cdot I, \quad R^u = 1.1315 \cdot I, \quad R^w = 120.2944 \cdot I.$

(a) Spectral radius $\rho(A - BK - CL)$       (b) Distance to NE

Figure 3: Stability issue (non-convergence) caused by bad initialization $K_0$ that is not robustly stabilizing. The iterations here denote the ones for updating $L$, with $K_0$ fixed.

We initialize $K_0$ as

$$K_0 = \begin{bmatrix} 2.1548 & 1.3041 & 0.7464 & -0.1457 \\ 1.5858 & 0.0962 & 1.7218 & 2.4581 \end{bmatrix},$$

which satisfies both $\rho(A - BK_0) = 0.9703 < 1$ and $\|\mathcal{T}(K_0)\|_\infty = 0.7718 < 1$, i.e., $K_0$ is robustly stabilizing. We choose $N_L = 1000$ and $N_K = 10^7$ (large enough), to perform the RARL scheme in [2]. We first fix $K_0$, and update $L_0 = 0$ to obtain $L_1$. As $\|\mathcal{T}(K_0)\|_\infty < 1$, $L(K_0)$ is well-defined, and has the value of

$$L(K_0) = \begin{bmatrix} -1.8588 & 0.4587 & -0.6982 & 0.1612 \\ -0.4100 & 0.0263 & -0.2527 & -0.1007 \end{bmatrix}.$$

This $L(K_0)$ can be achieved by $L_1$ within $N_L = 1000$ steps using the NPG update for $L$ with $\eta_L = 0.001$. This verifies the fast inner-loop convergence (linear rate) we established in Lemma 4.4.

(a) Spectral radius $\rho(A - BK - CL)$       (b) Distance to NE

Figure 4: Stability issue (non-convergence) due to bad choices of $(N_K, N_L)$. The iterations here denote the ones for updating $K$, with $L_1 = L(K_0)$ fixed.

Then, as in [2], with $L_1 = L(K_0)$ fixed, we improve $K$ from $K_0$ using the NPG update with stepsize $\eta = 10^{-9}$. However, as shown in Figure 4, such an infinitesimal stepsize still cannot prevent the system from destabilizing. This phenomenon is essentially due to that for fixed $L_1 = L(K_0)$, the inner-loop problem $\min_K \mathcal{C}(K, L_1)$ in turn is not necessarily well-defined. Indeed, Matlab returns a solvability issue for solving the following Riccati equation induced by $L_1 = L(K_0)$:

$$P_{L_1}^* = Q - L_1^\top R^w L_1 + (A - CL_1)^\top \left[ P_{L_1}^* - P_{L_1}^* B(R^u + B^\top P_{L_1}^* B)^{-1} B^\top P_{L_1}^* \right](A - CL_1),$$

where $P_L^* = P_{K(L),L}$ is the solution to the Lyapunov equation (2.5) under $(K(L), L)$. Hence, even with a robustly stabilizing initialization $K_0$, the RARL update can still destabilize the system easily if $N_K$ and $N_L$ are not chosen properly. This reinforces the stability issue we observed in **Example** 3.6.

(a) Distance to NE        (b) $\|P_{K,L} - P^*\|_2$

Figure 5: Convergence of the double-loop algorithm on 8 randomly generated examples.

The two stability issues highlight the significance of both the *initialization* (being robustly stabilizing), and the *update rule* (with properly chosen $(N_K, N_L)$). They motivate our robustification techniques in §5, and the policy-based RARL algorithms in §4, respectively.

## C.2    Results for Section §4: Double-Loop Algorithm

In the main paper, we have substantiated both the *stability* and *convergence* of the double loop algorithm in §4 theoretically. Now we verify our theory via simulation examples.

**Example C.1.** We select $A \in \mathbb{R}^{4 \times 4}$ to be $I + \xi$, where each element of $\xi \in \mathbb{R}^{4 \times 4} \sim \text{Unif}[0, 1]$; $B, C \in \mathbb{R}^{4 \times 2}$ with each element $\sim \text{Unif}[0, 1]$; $R^u, R^w \in \mathbb{R}^{2 \times 2}$ to be $I + \zeta \cdot I$, with $\zeta \sim \text{Unif}[0, 0.5]$; $Q$ to be $I + \varrho \cdot I$, with $\varrho \sim \text{Unif}[0, 0.1]$. We initialize $K_0$ randomly such that it satisfies the robust stability condition. For each fixed $K$, we perform NPG update (4.2) until it converges to $L(K)$. We randomly run 8 example cases, and plot the convergence of the distance to the NE, i.e., $\|K - K^*\|_2 + \|L - L^*\|_2$, and the norm of the difference between $P_{K,L}$ and $P^*$ in Figure 5. The stepsizes are $\eta = \eta_L = 0.0005$. It is shown that the double-loop algorithm converges to the NE as expected, without any projection as suggested in [33].

## C.3    Results for Section §4: Descent-Ascent Algorithms

Now we focus on the descent-ascent algorithms proposed in §4.4, the extensions of the double loop algorithm with finite $N_K$ and $N_L$ (without necessarily solving the inner loop to the optimum). These descent-ascent algorithms include the simultaneous-updating rule mentioned in §4.4. We present some simulation results that are not included in the main paper due to space limitation.

Our main findings are restated here for clarity. We show that in many cases, the descent-ascent updates with properly chosen $N_K$ and $N_L$ are effective, i.e., converge to the NE successfully. Most of the successful cases require the initialization $K_0$ to satisfy the robust stability condition. On the other hand, if the initialization is not robustly stabilizing, descent-ascent updates do not always work (but may work in some cases). These observations reinforce the significance of a *robustly stabilizing initialization*.

### C.3.1    Multi-Dimensional Convergent Cases

**Robustly Stabilizing Initialization**

We first test the descent-ascent NPG algorithms (with different choices of $N_K$ and $N_L$), when the initialization $K_0$ is robustly stabilizing, on the randomly generated cases in **Example** C.1. Recall that in **Example** C.1, we select $A \in \mathbb{R}^{4 \times 4}$ to be $I + \xi$, where each element of $\xi \in \mathbb{R}^{4 \times 4} \sim \text{Unif}[0, 1]$; $B, C \in \mathbb{R}^{4 \times 2}$ with each element $\sim \text{Unif}[0, 1]$; $R^u, R^w \in \mathbb{R}^{2 \times 2}$ to be $I + \zeta \cdot I$, with $\zeta \sim \text{Unif}[0, 0.5]$; $Q$ to be $I + \varrho \cdot I$, with $\varrho \sim \text{Unif}[0, 0.1]$. We use $(K_0, 0)$ as the initialization control gain pair. We randomly run 8 example cases, and plot the convergence of the distance to the NE, i.e., $\|K - K^*\|_2 + \|L - L^*\|_2$, and the norm of the difference between $P_{K,L}$ and $P^*$ in the following figures. The stepsizes are $\eta = \eta_L = 0.001$, and we choose $N_K = 1$ and $N_L$ ranging from $500, 50, 1$.

(a) Distance to NE             (b) $\|P_{K,L} - P^*\|_2$

Figure 6: Convergence of the NPG descent-ascent algorithm on $8$ randomly generated examples from **Example** C.1, with $N_K = 1$ and $N_L = 500$, and with robustly stabilizing initialization $K_0$. The iterations here denote the ones for updating $K$.

It is seen from Figures 6-8 that all three update rules converge to the NE successfully. In particular, the convergence with larger $N_L$ is relatively more stable, i.e., both the distance to NE and the norm difference $\|P_{K,L} - P^*\|_2$ converge to zero monotonically. For the case where $N_L = 1$, the convergence can be slow (see Figure 8 (a)), and the norm difference $\|P_{K,L} - P^*\|_2$ may not be monotonically decreasing (see Figure 8 (b)). But still, with robustly stabilizing initializations, these descent-ascent update rules with finite $N_L$, though not theoretically justified yet, are as effective as the double-loop update rule shown in Figure 5. We also observe that the stepsizes allowed to use here are a bit larger than those used in the double loop update rule that generated Figure 5. This yields a faster convergence, and an easier implementation without fine-tuning the stepsizes. All these positive observations have motivated us to study the convergence theory for the descent-ascent updates in our future work.

(a) Distance to NE             (b) $\|P_{K,L} - P^*\|_2$

Figure 7: Convergence of the NPG descent-ascent algorithm on $8$ randomly generated examples from **Example** C.1, with $N_K = 1$ and $N_L = 50$, and with robustly stabilizing initialization $K_0$. The iterations here denote the ones for updating $K$.

In addition, we also test the case with not only $N_K = N_L = 1$, but *simultaneously-updating* agents. Particularly, the update rule becomes

$$K' = K - \eta \nabla_K \mathcal{C}(K, L) \Sigma_{K,L}^{-1}, \qquad L' = L + \eta \nabla_L \mathcal{C}(K, L) \Sigma_{K,L}^{-1},$$

without any timescale separation between the two agents. This is much easier to implement than other double-loop/descent-ascent update rules. However, this seems to be less "safe" than the latter ones, and may easily oscillate since the simultaneous moving of the opponent makes the environment faced by one agent *less stationary*. Surprisingly, with our robustly stabilizing initialization, such seemingly non-stationary updates still work. As shown in Figure 9, simultaneously-updating scheme also converges to the NE for all $8$ examples. Here we use stepsizes $\eta = 0.001$ for both agents.

(a) Distance to NE

(b) $\|P_{K,L} - P^*\|_2$

Figure 8: Convergence of the NPG descent-ascent algorithm on $8$ randomly generated examples from **Example** C.1, with $N_K = 1$ and $N_L = 1$, and with robustly stabilizing initialization $K_0$. The iterations here denote the ones for updating $K$.

(a) Distance to NE

(b) $\|P_{K,L} - P^*\|_2$

Figure 9: Convergence of the NPG descent-ascent algorithm on $8$ randomly generated examples from **Example** C.1, with $N_K = 1$ and $N_L = 1$, and with robustly stabilizing initialization $K_0$. The iterations here denote the ones for updating both $K$ and $L$, as they are updated simultaneously.

(a) Distance to NE

(b) $\|P_{K,L} - P^*\|_2$

Figure 10: Convergence of the NPG descent-ascent algorithm on $8$ randomly generated examples from **Example** C.1, with $N_K = 1$ and $N_L = 1$, and with *non-robustly stabilizing* initialization $K_0$. The iterations here denote the ones for updating $K$.

Similar behaviors occur as in the descent-ascent algorithm with $N_K = N_L = 1$ (see Figure 8), i.e., slower convergence can happen in some examples, and the norm difference $\|P_{K,L} - P^*\|_2$ is not monotonically decreasing.

**Non-Robustly Stabilizing Initialization**

Interestingly, when both $N_K$ and $N_L$ are finite, sometimes even a non-robustly stabilizing $K_0$ would work. Consider the same setting as above (also as in **Example** C.1), except that the matrix $R^w$ is now chosen differently. In order to easily generate initialization $K_0$ that is *stabilizing* but not *robustly stabilizing* for certain $R^w$, we here first generate $K_0 \in \mathbb{R}^{4 \times 2}$ randomly, with each element $\sim \text{Unif}[-3.5, 3.5]$ such that $\rho(A - BK_0) < 1$. We then calculate the $\mathcal{H}_\infty$-norm $\|\mathcal{T}(K_0)\|_\infty$, with $\mathcal{T}(K_0)$ defined in (5.1), and $R^w$ therein replaced by $I$. Then, we choose $R^w = \tau \cdot \|\mathcal{T}(K_0)\|_\infty \cdot I$ for some $\tau \in (0, 1)$. This way, the actual $\mathcal{H}_\infty$-norm $\|\mathcal{T}(K_0)\|_\infty$ under the chosen $R^w$ will be $> 1$, i.e., $K_0$ is non-robustly stabilizing. We have randomly generated 8 examples with $\tau \sim \text{Unif}(0.7, 1)$, and tested the descent-ascent update with $N_K = N_L = 1$ (the most "non-stationary" one according to Figures 6-8). We choose the stepsizes $\eta = \eta_L = 0.001$. As shown in Figure 10, the descent-ascent algorithm still converges successfully to the NE.

This implies that the descent-ascent algorithms with a carefully chosen $(N_K, N_L)$ may have the potential to *enlarge the convergent initialization*, compared to the double-loop counterparts. Recall that for the latter, a non-robustly stabilizing $K_0$ is not even sensible, since the inner-loop problem over $L$ is not even well-defined for this $K_0$. Thus, the inner-loop update will blow up with a large $N_L$ (and even an infinitesimal stepsize) (see **Example** 3.2 and §C.1.1). This can be more desired in practice, as it shows that the requirement for initialization can be relaxed sometimes, for descent-ascent updates. However, lacking theoretical guarantees, it may still be unsafe to use non-robustly stabilizing initializations, as we will show next in §C.3.2.

### C.3.2   Multi-Dimensional Non-Convergent Cases

It is not uncommon to have cases where a *non-robustly stabilizing* initialization yields non-convergent results for descent-ascent algorithms. We enumerate four such randomly generated cases with different random seeds. The examples are also generated as in **Example** C.1, except that as above, the non-robustly stabilizing initialization is produced by first randomly generating a stabilizing $K_0$, and then select $R^w = \tau \cdot \|\mathcal{T}(K_0)\|_\infty \cdot I$ for some $\tau \in (0, 1)$. We choose $\tau = 0.3$ for Case 1, and $\tau = 0.05$ for Cases $2, 3, 4$. The first two cases use the same random seed. We choose the stepsizes $\eta = \eta_L = 10^{-6}$, much smaller than those used for the convergent cases in §C.3.1. We also choose $N_L = 60$, a moderate number.

|(a) Case 1|(b) Case 2|(c) Case 3|(d) Case 4|

Figure 11: Non-convergence of descent-ascent algorithms due to non-robustly stabilizing initialization. After large enough numbers of iterations, even with an infinitesimal stepsize, the spectral radius $\rho(A - BK - CL)$ still blows up to $> 1$. The iterations here denote the ones for updating $K$.

As shown in Figure 11, the descent-ascent algorithm drives the spectral radius $\rho(A - BK - CL)$ to be $> 1$, and finally destabilizes the system, for all four cases. This implies that the success in Figure 10 is only because $\tau$ is relatively large ($\tau \sim \text{Unif}(0.7, 1)$), i.e., $K_0$ is *not too far* out of the robustly stabilizing region. With a much less robust initialization $K_0$ when $\tau$ is small, the descent-ascent algorithms can still fail. Interestingly, note that Cases 1 and 2 have the same random seed, and the only difference is that the $\tau$ for Case 1 is larger, i.e., the initialization $K_0$ is *less robust* under the $R^w$ in Case 2. Hence, intuitively, the algorithm should explode faster in Case 2. However, the more robust initialization in Case 1 enables a much faster explosion (as per Figure 11 (a) and (b)). This implies that it is not necessarily true that the farther away from the robustly stabilizing region, the faster it will destabilize the system. This highlights the complexity of the optimization landscape of policy-based LQ RARL. We further illustrate this by revisiting the one-dimensional case in §3 next.

### C.3.3   Revisiting the One-Dimensional Example

We now revisit the one-dimensional case in **Example** 3.2. As shown in (12), we recall that the blue, cyan, and green regions represent the stabilizing region of $(K, L)$ (where $\rho(A - BK - CL) < 1$),

Figure 12: Illustrating the region of initializations of $(K, L)$ (in black) that yield the convergence of the descent-ascent algorithms, with different $N_L$, on the one-dimensional case in **Example** 3.2.

the stabilizing region of $K$ (where $\rho(A - BK) < 1$), and the robustly stabilizing region of $K$ (where $\|\mathcal{T}(K)\|_\infty < 1$), respectively. The former region includes the latter ones. The Nash equilibrium of the game (A.8) (in red), which yields the optimal robust controller, is contained in the robustly stabilizing region.

We then test the convergence of the descent-ascent algorithms. For the first case, we choose $N_K = N_L = 1$, and stepsizes $\eta = \eta_L = 0.05$. We label all the initializations that yield convergence to the NE, named *convergent initialization*, as black in Figure 12 (a). Recall that for the double loop algorithm, where $N_L \to \infty$, the convergent initialization region has been proved to be the robustly stabilizing region (green) (see Theorem 4.2). Indeed, for those $K$ outside the green region, the inner-loop optimization over $L$ is not even well-defined. Interestingly, it is seen in Figure 12 (a) that the convergent initialization region is *enlarged* by the descent-ascent algorithm, as we have observed in the multi-dimensional cases in §C.3.1. Such an enlarged region contains the green one, and even reaches the $(K, L)$ stabilizing region where $K$ is not stabilizing (blue region). This might be appealing in practice, as a stabilizing $K$ (easier to obtain than a robustly stabilizing one) might be enough for initialization. But still, as we point out in the multi-dimensional case in (C.3.2), this may not be safe, since there are other parts in cyan that do not lead to convergence. Careful choices of $N_L$, $\eta$, and $\eta_L$ are crucial for the update rule to work.

We have also increased $N_L$ to 30, and used stepsizes $\eta = 0.05$ and $\eta_L = 0.01$. As shown in Figure 12 (b), the convergent initialization region is smaller than that in (b), but still goes beyond the robustly stabilizing region (green). This means that the setup with $N_K = N_L = 1$, though might be less stable *during* the iterations (see Figures 8 and 10), requires less restricted initializations than those with $N_L = 30$. Another interesting observation is that there is some small green region that *does not* lead to convergence. This shows that even a robustly stabilizing initialization may not be effective for descent-ascent algorithms, with certain choices of the stepsizes and $(N_K, N_L)$. This reaffirms the complicated intertwinement between the *update rule* and the *initialization*, in order to guarantee robust stability on-the-fly.

In sum, the observations in this subsection have highlighted the significance of a robustly stabilizing initialization: it is the safest one that enables all proposed RARL schemes to work, though it can be relaxed sometimes with a carefully chosen $(N_K, N_L)$, and an initialization not too far out of the robustly stabilizing region.

## C.4 Results for Section §5

In this section, we validate the effectiveness of our $\mathcal{H}_\infty$-based *initialization robustification* techniques proposed in §5.

### C.4.1 Finite-Difference Method

We first test the finite-difference method. As in the one-dimensional example, i.e., **Example** 5.4, in §5, we robustify the case that has stability issues caused by a bad initialization, i.e., the case in §C.1.1. Recall the robustification update rule in (5.2), and the finite-difference estimate of the subgradient of $\|\mathcal{T}(K)\|_\infty$ in (5.3). Here we choose $\epsilon = 10^{-10}$ in (5.3), and the stepsize $\alpha = 5 \times 10^{-9}$ in (5.2). We require the robustified initialization to have the $\mathcal{H}_\infty$-norm $\|\mathcal{T}(K)\|_\infty < 0.9$.

As shown in Figure 13 (a), $\|\mathcal{T}(K)\|_\infty$ decreases monotonically to be $< 1$ (in fact, $< 0.9$) using the simple update (5.2), within only 9 iterations. After the robustification, as shown in (b) and (c), all three proposed RARL schemes converge to the NE, in contrast to the divergence shown in Figure 3.

(a) Robustification using (5.2)  (b) Distance to NE  (c) $\|P_{K,L} - P^*\|_2$

Figure 13: Illustrating the effectiveness of the $\mathcal{H}_\infty$-based robustification using the finite-difference method, on the non-convergent case in §C.1.1. (a) shows the convergence of the update rule (5.2). (b) and (c) show the convergence of all proposed RARL schemes after the robustification. The iterations here denote the ones for updating $K$.

In addition, we have also tested the method on the first non-convergent case, Case 1, in §C.3.2. As shown in Figure 14, the robustification technique still works, and enables all three RARL schemes to converge to the NE successfully, including the descent-ascent algorithm with $N_L = 60$, which did not work before robustification, as shown in §C.3.2.

(a) Robustification using (5.2)  (b) Distance to NE  (c) $\|P_{K,L} - P^*\|_2$

Figure 14: Illustrating the effectiveness of the $\mathcal{H}_\infty$-based robustification using the finite-difference method, on the first non-convergent case, Case 1, in §C.3.2. (a) shows the convergence of the update rule (5.2). (b) and (c) show the convergence of all proposed RARL schemes after the robustification. The iterations here denote the ones for updating $K$.

### C.4.2 Stochastic Zeroth-Order Method

In addition to the finite-difference method, the subgradient of $\|\mathcal{T}(K)\|_\infty$ can also be estimated via *stochastic zeroth-order* methods [59, 60, 61], which can be viewed as stochastic variants of the finite-difference method. Specifically, the subgradient is estimated by $\widehat{g} \in \mathbb{R}^{m_1 \times d}$, which is defined as

$$\widehat{g}(K, \Delta) = \widehat{g} := \frac{\widehat{\|\mathcal{T}(\widetilde{K})\|_\infty} m_1 d}{r} \cdot \Delta, \qquad \text{with} \quad \widetilde{K} = K + r\Delta, \qquad \text{(C.1)}$$

(a) Robustification using (5.2)     (b) Distance to NE     (c) $\|P_{K,L} - P^*\|_2$

Figure 15: Illustrating the effectiveness of the $\mathcal{H}_\infty$-based robustification using the one-point stochastic zeroth-order method, on the non-convergent case in §C.1.1. (a) shows the convergence of the update rule (5.2), when (C.1) is used for subgradient estimation. (b) and (c) show the convergence of all proposed RARL schemes after the robustification. The iterations denote the ones for updating $K$.

where $\widehat{\|\mathcal{T}(\widetilde{K})\|_\infty}$ is a noisy stochastic estimate of the actual $\mathcal{H}_\infty$-norm $\|\mathcal{T}(\widetilde{K})\|_\infty$, $r > 0$ is the *smoothing radius*, and $\Delta \sim \mathrm{Unif}(\mathbb{S})$, with $\mathbb{S} := \{\Delta \in \mathbb{R}^{m_1 \times d} : \|\Delta\|_F = 1\}$ being the sphere of matrices with Frobenius norm 1. Note that if $\widehat{\|\mathcal{T}(\widetilde{K})\|_\infty}$ is an unbiased estimate of $\|\mathcal{T}(\widetilde{K})\|_\infty$, and independent of $\Delta$, then $\widehat{g}$ is an unbiased estimate of the subgradient of $\|\mathcal{T}(K)\|_\infty$. This estimate in (C.1) is also referred to as *one-point* estimate [59, 61]. Another common estimate is the *two-point* one [62, 61], where

$$\widehat{g}(K, \Delta) = \widehat{g} := \frac{\left[\widehat{\|\mathcal{T}(\widetilde{K}_1)\|_\infty} - \widehat{\|\mathcal{T}(\widetilde{K}_2)\|_\infty}\right] m_1 d}{2r} \cdot \Delta, \quad \text{with} \quad \widetilde{K}_1 = K + r\Delta, \quad \widetilde{K}_2 = K - r\Delta. \tag{C.2}$$

Both approximations become more and more accurate as the smoothing radius $r$ gets smaller. Nonetheless, small values of $r$ may lead to estimates with large variances. Also, such an $r$ should not be too large, as it needs to ensure that $\widetilde{K}$ is stabilizing (so that $\|\mathcal{T}(\widetilde{K})\|_\infty$ is well-defined). An appropriate choice of $r$ is thus critical for the algorithm to perform well. With these approximations at hand, the update becomes $K' = K - \alpha\widehat{g}$ with some stepsize $\alpha > 0$.

Compared with the finite-difference method, in particular, the subgradient does not need to be estimated element-wise. Moreover, the oracle that calculates the $\mathcal{H}_\infty$-norm needs not to be *exact*. It can be a stochastic and noisy estimate of it.

We use both one-point and two-point zeroth order methods to robustify the non-convergent case in §C.1.1. Here we choose $r = 10^{-4}$, $\alpha = 5 \times 10^{-10}$ and $\alpha = 10^{-9}$ for one-point and two-point updates, respectively. Also, we allow some noises to be added onto the actual $\|\mathcal{T}(\widetilde{K})\|_\infty$, i.e., $\widehat{\|\mathcal{T}(\widetilde{K})\|_\infty} = \|\mathcal{T}(\widetilde{K})\|_\infty + \sigma$, with $\sigma$ being drawn from standard normal distribution $\mathcal{N}(0, 1)$. As in §C.4.1, we require the robustified $K_0$ to have $\|\mathcal{T}(K_0)\|_\infty < 0.9$.

As shown in Figures 15 and 2, both robustification updates indeed reduce $\|\mathcal{T}(K)\|_\infty$ to be $< 0.9$, within 4 and 10 iterations, respectively. Note that the one-point method usually suffers from larger variances, requiring a smaller stepsize $\alpha$. Indeed, the $\mathcal{H}_\infty$-norm does not decrease monotonically in Figure 15 (a). Convergence of the two-point method is more stable, even with a larger stepsize $\alpha$. However, as our objective here is only to decrease $\|\mathcal{T}(K)\|_\infty$ to be $< 0.9$, the advantages of the two-point method may not be very essential. One-point method may robustify $K_0$ even faster. With a robustified initialization in hand, all three RARL schemes enjoy similar convergence performance, as that obtained using the finite-difference method, as shown in Figure 13.

Finally, we point out that although the method seems quite effective, and inherits the idea from zeroth-order smooth (convex) optimization, its global convergence can be challenging to analyze. Indeed, establishing the convergence of zeroth-order methods for nonconvex-nonsmooth optimization is still a relatively open and active research area [68, 69, 70, 71], wherein most of the works only address specific settings such as the objective with a finite-sum form and/or with only nonsmooth regularization terms. We leave the global convergence analysis of this approach as future work.