[Reviews · NeurIPS 2020]

Review 1

Summary and Contributions: The authors proposed a method for designing robust-adversarial LQR controllers using a double-loop policy gradient ascent/descent method. The main contribution is that the controller is designed to be robustly stabilizing during adaptation, which is very necessary for practical applications.

Strengths: The paper is clear and its novelty is in a direction well-aligned with NeurIPS community. Assumptions make sense, and claims are sound. Good empirical evidence is provided. Examples are illustrative and excellent for understanding. Great paper.

Weaknesses: Two limitations: (1) globally linear dynamics are rare in practice: this reviewer would like to know whether this could potentially be used for systems with local linear dynamics and what the problems arising in such scenarios could be? and (2) I believe the authors know the system matrices and description etc. Can then classical robust control methods be used to come up with LQ policies rather than RARL? Also, authors should mention that this is model-based approach.

Correctness: I believe so. Code is presented and looks fine.

Clarity: Yes, I had little to no trouble understanding the main paper. The supplementary is a bit haphazard and could benefit from a table of contents or organizational paragraph.

Relation to Prior Work: Reminiscent of this work, a double-loop approach to LQ control with state and input constraint satisfaction has recently been proposed where the authors are also keen to ensure recursive feasibility and stability properties: [A] https://arxiv.org/pdf/1906.11369.pdf. The current paper deals with the robust/disturbance case but some of the data-driven transformation of model-based optimization problems considered in [A] might be interesting to the authors. There is also work in the approximate dynamic programming literature by Frank Lewis and others in robust/H-infinity RL for linear systems - this could be relevant to the current paper too perhaps.

Reproducibility: Yes

Additional Feedback: - Which matrices are known at design time? Is this model-based RL? This needs clarification early on e.g. after 2.1 - Is there a reason the authors use min sup instead of min max or inf sup? --- POST REBUTTAL --- My opinion remains the same. Good paper, authors rebuttal improved things even further. - Equation (2.5) is not a Lyapunov equation (A'PA - P = -Q is). This is closer to a Riccati equation. - L. 136-137 For self-containment of the paper, please add how the gradients are computed, briefly. - In the thought experiment, there is no need to assume a K exists. If (A,B) is controllable and (A, Q^1/2) is observable for psd Q, such a K exists for L=0.


Review 2

Summary and Contributions: The paper extends the analysis of RARL (Robust Adversarial RL), an algorithm used to train robust policies in RL, to the state-feedback control of discrete-time dynamical systems (specifically, the discrete LQR case). The authors first discuss some stability issues that can arise from blindly applying RARL to dynamical systems, and help the reader better understand how to use RARL. Then, the authors provide an algorithm that has stability and convergence guarantees. An extensive numerical section is provided in the appendix.

Strengths: I believe the paper to be clear and well written. The claims seem sound, and the results provided in the paper shed a light on the applicability of RARL. The various results presented in the paper, together with the numerical section, greatly help with the understanding of the method.

Weaknesses: - A proper conclusion is missing. - The related work section can be improved. The authors can also better explain the difference with the work in [10] -Unfortunately the paper analyses only state-feedback control. It would be nice to see also an analysis of output feedback control. -Theoretically speaking, LQR is a good algorithm to analyse. Despite that, it is almost never used in practice (from my experience). It is hard to specify design requirements using LQR, especially in the frequency domain. It would be nice to analyse RARL in case there are some control requirements.

Correctness: The method seems correct.

Clarity: I believe the paper to be clear and well written

Relation to Prior Work: Relation to prior work is discussed, though can be made clearer.

Reproducibility: Yes

Additional Feedback: - How did you choose the A,B,C matrices in the numerical sections? == Post rebuttal I have read the authors rebuttal. I will not change my score.


Review 3

Summary and Contributions: The authors consider the Robust Adversarial Reinforcement Learning (RARL) approach in the Linear Quadratic (LQ) setting and investigate the stabilization capabilities of the resulting linear static feedback controller. First the setting, similar to e.g. [44], is described and relevant concepts are introduced and connections to related work are pointed out. As a first contribution the authors demonstrate with several examples, using numerical simulations, that RARL in the LQ setting can suffer from stability problems. One source of these problems are bad initializations of the policy, and the authors propose a simple criterion to avoid such problems. Another source of stability problems is a bad choice of the number of iterations performed in the algorithm. Next, the authors propose a new concrete RARL algorithm for the LQ setting and provide convergence and stability results for the algorithm, as well as numerical experiments demonstrating the algorithm on simple examples. Furthermore, two variants of the algorithm are presented and investigated with numerical examples, though no theoretical results are provided in these cases. Finally, a new robustification method based on robust control techniques is described, that is used to generate suitable initializations of the RARL algorithms. Overall, the paper makes a solid technical contribution. The findings, methods and theoretical results are interesting and relevant. However, the authors consider a rather specific approach (general approach from [30], setting from [44] and [10]) and the methods and theoretical findings appear somewhat incremental rather than novel or a break-through. While this work definitely should be published, it is questionable whether the contributions are considered sufficiently novel for NeurIPS.

Strengths: * The authors demonstrated stability issues of RARL in the LQ setting. Since RARL seems to be popular and the LQ setting is of fundamental importance (in particular as a benchmark), this is a relevant contribution to the reinforcement learning community. * The authors propose a simple concept for good policy initializations in order to avoid stability problems (Def 3.3). This is then used later on to provide stability guarantees for a new RARL algorithm. A strength of this approach is its conceptual simplicity and the clear connection to established robust control theory (made explicit in Lemma 3.4) * A new, simple RARL algorithm for the LQ setting is proposed (Section 4.1) with stability and convergence guarantees (Sections 4.2, 4.3). The algorithm itself seems to be just a minor extension of prior work, in particular [44], however, the particular stability and convergence guarantees seem to be new, and hence this can be viewed as a relevant contribution. The techniques used for the theoretical results are standard, though one could argue that this is actually a strength, since the authors demonstrate that established methods can be successfully used to tackle problems of reinforcement learning. According to the authors similar results are derived in [10], but the latter work contains a technical error (and uses a different technique anyway), details are provided in Remark B.1 in the supplementary. * All claims in the paper are supported also by numerical experiments on synthetic examples. It seems that all the necessary Matlab code for reproducing all experiments is provided (though I haven't run all the scripts). The code is rather scarcely documented, but it is enough to understand everything.

Weaknesses: * Only sublinear convergence can be guaranteed for the algorithm proposed in Section 4.1 It would have been nice if the authors discussed this in more detail, in particular also reporting the run times of the numerical experiments and discussing the current limits imposed by the convergence rates. Note that in Remark 4.3 and Section B.4 faster (local) convergence rates are discussed. However, as noted by the authors acceleration with Gauss-Newton method cannot be done in a model-free manner. Since this paper is aimed at the reinforcement learning community this seems to be a major weakness. This aspect makes the discussion of the slow guaranteed convergence mentioned above even more important. * The discussion of the stability problems in Section 3.2 is very vague. In particular, it is not clear how Example 3.5 is related to the (very short) discussion preceding it. Reading Section C.1.2 makes things a bit clearer, but Section 3.2 should be improved. Furthermore, it would be nice if Section 3 contains a summary like in lines 718-720 in the supplementary.

Correctness: * Theory: The theory in Section 2, 3 and Section A in the supplementary is standard (basic robust control theory and game theory, some recent results from [44]) and hence can be assumed to be correct. New theoretical contributions are in Section 4 and Section B in the supplementary and seem to be correct (though I have not recalculated every step). * Empirical results: All experimental setups are reasonable and seem to be correctly implemented and the results transparently reported. However, it would have been nice to get some information on runtime as well as the underlying standard methods used in Matlab. * In lines 13-15 the authors claim "We explain the difference between robust stability and robust performance, and then discuss several ways to learn robustly stabilizing initializations". This statement seems misleading since robust performance (in the sense of robust control) is not discussed at all (in fact, the term robust performance does not appear outside the abstract and introduction at all).

Clarity: * The theoretical contributions (including proofs) as well as the experiments (including results) are clearly described. * While the technical contributions are clearly written, in general the paper is a bit difficult to read. In particular the Introduction seems to lack a clear structure and Section 2 is difficult to understand without reading Section A in the supplementary or being already familiar with this particular line of work. Interestingly, while the paper indeed tells a coherent story (problems of RARL in LQ setting -> approach to overcome this -> theoretical guarantees) this is not reflected in the text and it feels like a collection of different pieces. Again, this is an issue with the writing and not with the content. * Furthermore, it would have been nice to have a proper Conclusion (in fact the authors put a short conclusion in the Broader Impact section)

Relation to Prior Work: * The authors state that RARL [30] is popular, however, there are only few references to substantiate this claim. * The connection to reinforcement learning (in particular [30]) and previous work (mostly [10], [44]) is clearly described. However, the related work section is very short. In particular, a brief description of other recent usages robust control methods and theory for reinforcement learning would have been nice. Furthermore, although many references for estimating the $H_\infty$ norm from data are provided in Section 5, almost no pointers to other approaches of model-free robust control methods are given (there has been quite some activity, cf. e.g. [R1]) * In lines 331-332 the authors write that "We note that such a model-free robustification approach seems novel even in the robust control area, to the best of our knowledge." This could potentially be a bit misleading since model-free approaches for robust controller synthesis (which can be interpreted as a form of robustification) are subject of recent investigation, cf. e.g. [R2] for an LMI-based approach and [R3] for MPC approaches. However, I would consider this only a minor issue. [R1] De Persis, Claudio, and Pietro Tesi. "Formulas for data-driven control: Stabilization, optimality, and robustness." IEEE Transactions on Automatic Control 65.3 (2019): 909-924. [R2] Berberich, Julian, et al. "Robust data-driven state-feedback design." arXiv preprint arXiv:1909.04314 (2019). [R3] Berberich, Julian, et al. "Data-driven model predictive control with stability and robustness guarantees." IEEE Transactions on Automatic Control (2020).

Reproducibility: Yes

Additional Feedback: * More details should be provided for Lemma 3.4 and Lemma 5.2, at least the concrete version of the Bounded Real Lemma used (Reference [45] has many versions) Minor language issues * l40: "on solid grounding" instead of "under" * l41: "robustness performance" unclear, probably robust performance is meant * l147: "if it exists" and "be worsened" * The paragraph starting from l153 should not be called a thought experiment. It is more like a discussion. ================================= POST REBUTTAL Thank you for the clarifications, esp. wrt the contributions and relations to [10], [44], which was convincing. In light of the rebuttal and discussion, I've increased my score +1.


Review 4

Summary and Contributions: This paper studies robust adversarial RL (RARL) under LQ setting. It argues that conventional RARL scheme that greedily alternates agents’ updates can easily destabilize the system. Authors propose an algorithm to resolve such issue.

Strengths: This paper discusses the stability issues in policy gradient updates of LQ RARL. Authors propose a double loop algorithm to achieve stability.

Weaknesses: This work is limited in linear quadratic case, and PG under LQ case is well studied. So the technical challenges are not clear.

Correctness: The method seems correct. There is no empirical study in the main paper.

Clarity: This writing is understandable, however more details such as assumptions should be discussed more clearly.

Relation to Prior Work: No, this paper mentions very limited related work.

Reproducibility: Yes

Additional Feedback: Overall, I have a bit negative opinion of the paper. My main concerns include: 1, the related work is not well discussed. 2, the technical contribution is not clear, especially for LQ case. 3, the assumptions are not well discussed. Authors define robust stability condition, which essentially makes a critical intermediate term in analysis easy to deal with. A more reasonable assumption should be imposed on A,B,C. 4, the necessity of robust stability condition is not rigorously proved. == Post rebuttal 1, some potential references about RARL that should be included are: Extending robust adversarial reinforcement learning considering adaptation and diversity, Shioya et al 2018; Adversarial Reinforcement Learning-based Robust Access Point Coordination Against Uncoordinated Interference, Kihira et al 2020; Robust multi-agent reinforcement learning via minimax deep deterministic policy gradient, Li et al 2019; Policy-Gradient Algorithms Have No Guarantees of Convergence in Linear Quadratic Games, Mazumdar et al 2019; Policy Iteration for Linear Quadratic Games With Stochastic Parameters, Gravell et al 2020; Risk averse robust adversarial reinforcement learning, Pan et al 2019; Online robust policy learning in the presence of unknown adversaries, Havens et al 2018. 2, I did not read supplementary material so I neglect the simulations provided by authors. It seems authors conducted multiple experiments to support the theory. 3, The explanation of assumption is not satisfactory. First, the assumption (Assump A.1) is not included in the main paper, but is cited. Second, authors argued Assump A.1 is weaker than imposing assumption on A,B,C and referred to [5, Chapter 3]. I looked over this reference and could not find such argument. Overall, I'm persuaded by the author's response regarding the experiments and the interest of the problem (though I personally don't like the time-invariant LQ setup). But I still have a bit negative opinion regarding the presentation and related work section.

[Author Response · NeurIPS 2020]

**Rev 1.** We thank the reviewer for the positive feedback. (1) On local linear models: One possible approach is to apply RARL to each local linear model and then use the idea in guided policy search to piece together all local controllers. One issue is that there will not be a "global" robustness guarantee. But our results still hold locally. (2) Model-based or model-free?: With model-knowledge, other methods from classical robust-control, such as LMIs or Riccati equations can be used. However, these methods can hardly be made "model-free", and are less scalable (to high dim. systems) than our PG methods (which can also be made model-free via zeroth-order optimization methods). We assume here that the model is known, since our focus is on the fundamental issues regarding "optimization landscape" and "stability" in LQ RARL: model-based update already illustrates the landscape well; while sample-based update will only worsen the stability issue we've identified. We have mentioned this in lines 137-140, and will emphasize it in revision. (3) We will include the references. (4) The reviewer's suggestions on improvements are helpful. We will revise accordingly.

**Rev 2.** We thank the reviewer for the positive feedback. With an extra page allowed, we will be able to add conclusion and more details on related work. Regarding the restrictions of the state-feedback case, we agree that the output feedback case is important. The "optimization landscape" might become more challenging for the output-feedback case, and there has been little theoretical work even on "PG for output-feedback (non-robust) LQG". We leave this as an important future direction. Regarding the control requirements, in the robust control context we have studied, the frequency domain requirement on "$\mathcal{H}_\infty$-norm" is equivalent to some LMI or Riccati conditions in the time domain (see our Lemmas 3.4 and 5.2), and can thus be imposed. Imposing *other* control requirements in RARL is left for future research. On $(A, B, C)$, we chose the matrices randomly, and made sure that they satisfy our assumptions.

**Rev 3.** We appreciate the detailed and positive comments, and hope that our response below addresses your concerns and help improve the scoring. (1) *Novelty:* We respectfully disagree that our contributions are incremental: (a) RARL in [30] is a highly-cited approach, and the stability issue of RARL that we've identified has been overlooked in (RA)RL; (b) the convergence theory and proof techniques are different from either [44] or [10]. Our assumptions are also different from [10] and [44], and align much better with the common assumptions in robust control. We note that even in the zero-sum LQ game context, our proofs are the first correct ones that carry rigorous *robust control* implications; (c) Our algorithms are not "minor extensions" of those in [44]. "(Robust) stability" has been handled in [44] through a rough "projection" step, which requires *model-knowledge*, and has no robust control implications; we had new and non-trivial techniques to remove the projection, enabling model-free algorithm-design; (d) empirically, the study of other descent-ascent PG methods, and how the joint effect of "initialization" and "update-rule" affects the convergence is new, while [10] did not provide any empirical results or any study on descent-ascent methods; (e) the "robust control implication" of the "good initializations" in LQ RARL (satisfying certain $\mathcal{H}_\infty$-norm constraint), and the "zeroth-order optimization-based" robustification algorithm are both novel, and cannot be found in either the (RA)RL or control literature (including [10,30,44]). (2) Convergence rates: Yes, only sublinear rates were established. For nonconvex optimization without additional problem structure, e.g., the gradient domination property of the objective for the inner-loop LQR, this *global sublinear* rate is something one can hardly improve in general. But note that in our simulations (Figure 4), convergence of this double-loop algorithm is not that bad (sublinear only in the beginning). We will add the runtime discussions. Also, note that the discussion in Sec. B.4 about faster local linear rates is not only for *Gauss-Newton (G-N)*, but also for *natural PG*, which can be made model-free using zeroth-order methods. Finally, we would like to clarify that when saying "G-N cannot be made model-free", we meant that the "zeroth-order optimization"-based methods cannot be used for G-N. It is still possible to apply other methods. For the (non-robust) LQR problem, the G-N method can be implemented in a model-free manner using the approximate policy iteration method where the policy evaluation step uses LSTD-$Q$. The RARL case is similar. We shouldn't have made it sound like a dead-end. We will add clarifications. (3) On stability issues in Sec. 3.2: Example 3.5 is a "best-response" update (with large enough $N_K, N_L$), which was discussed in the paragraph before it. We will add more details like lines 718-720. (4) Suggestions on clarity: We will revise accordingly. (5) Ref.: [30] is indeed highly-cited and we will add evidence. Thanks for mentioning the other references. We will include them. [R2] is very helpful for robustification.

**Rev 4.** Thanks for the comments. We hope our response will help for re-evaluating our work. (1) "Limited work": We respectfully disagree that our work is "limited". It is well-acknowledged that, LQ is the most fundamental and common setting in (robust) control, covering also scenarios where nonlinear, norm-bounded perturbations are allowed around a nominal linear system. To our knowledge, "PG for LQ setting" has only been well-studied for "non-robust LQR" problems, *but not* for zero-sum LQ games, or robust control. The most related work [10,44] has already been discussed in detail. See reply (1) to **Rev. 3**. We have novel technical improvement over [10,44]. (2) "No empirical study in the main paper": Figure 1 is an empirical result, and more results are available in the appendix, with clear pointers in the main paper. (3) "Limited related work": Could the reviewer specify which references? (4) Reproducibility: We have provided all the code and experiment details, and all other reviewers see our work as reproducible. Could the reviewer specify it is not reproducible in what sense? (5) *Assumptions:* Robust stability is not an assumption. Guaranteeing it in fact makes the analysis harder. Assump. A.1 on the existence of the solution to GARE is standard in robust control (cf. [2,5,37]), and is weaker than the direct assumption on $A, B, C$, see [5, Chapter 3]. Robust stability is a significant property in robust control, and is essential in our LQ RARL. It is not some "necessary condition".

[Meta-Review · NeurIPS 2020]

This paper studies a recent method on Robust Adversarial Reinforcement Learning (RARL) by Pinto et al in the linear quadratic setting (linear dynamics, quadratic cost function), which is a typical starting point in the analysis of optimal control algorithms. The paper examines the stabilization behavior of the linear controller, showing that RARL in the simplified linear quadratic setting shows instabilities. The paper proposes a new formulation of RARL in the linear quadratic setting, which can inform solutions in the nonlinear setting, and provides stability guarantees for the proposed method. In the post rebuttal discussion 3/4 reviewers evaluated the paper highly and recommended that the paper be accepted. 2/4 reviewers expressed concerns about the limited novelty of the paper, as it relies heavily on [30, 44, 10]. I agree that the paper makes a significant and interesting enough contribution in terms of pointing out the instabilities of RARL and addressing them in the linear quadratic setting, which in my view is sufficient for publication at NeurIPS.